# Surface enhanced Raman scattering artificial nose for high dimensionality fingerprinting

Nayoung Kim [1,4], Michael R. Thomas[1,4], Mads S. Bergholt[1], Isaac J. Pence [1], Hyejeong Seong [1], Patrick Charchar [2], Nevena Todorova [2], Anika Nagelkerke[1], Alexis Belessiotis-Richards [1], David J. Payne [3], Amy Gelmi[1], Irene Yarovsky [2]* & Molly M. Stevens [1]*

Label-free surface-enhanced Raman spectroscopy (SERS) can interrogate systems by directly fingerprinting their components' unique physicochemical properties. In complex biological systems however, this can yield highly overlapping spectra that hinder sample identification. Here, we present an artificial-nose inspired SERS fingerprinting approach where spectral data is obtained as a function of sensor surface chemical functionality. Supported by molecular dynamics modeling, we show that mildly selective self-assembled monolayers can influence the strength and configuration in which analytes interact with plasmonic surfaces, diversifying the resulting SERS fingerprints. Since each sensor generates a modulated signature, the implicit value of increasing the dimensionality of datasets is shown using cell lysates for all possible combinations of up to 9 fingerprints. Reliable improvements in mean discriminatory accuracy towards 100% are achieved with each additional surface functionality. This arrayed label-free platform illustrates the wide-ranging potential of high-dimensionality artificial-nose based sensing systems for more reliable assessment of complex biological matrices.

---

[1] Department of Materials, Department of Bioengineering and Institute of Biomedical Engineering, Imperial College London, London SW7 2AZ, UK. [2] School of Engineering, RMIT University, Melbourne, Victoria, Australia. [3] Department of Materials, Imperial College London, London SW7 2AZ, UK. [4] These authors contributed equally: Nayoung Kim, Michael R. Thomas *email: irene.yarovsky@rmit.edu.a; m.stevens@imperial.ac.uk

There are huge benefits to be had in moving towards platform diagnostic technologies that are not reliant on target-specific binding structures (antibodies, aptamers etc.) and the associated burden of their discovery, complex conjugation and production procedures. A number of targeting-free sensing technologies are being developed that seek to meet this goal[1–3], and among them, label-free surface-enhanced Raman spectroscopy (SERS) has attracted considerable attention with the promise of sensitive direct profiling of the unique fingerprints of a biological sample in a wash- and label-free format[4–6]. Compared with a targeted approach of detecting the presence of a specific analyte by measuring signals from pre-tagged Raman reporters, a key advantage of label-free SERS is that it is not necessarily limited to pre-specification of targets of interest and the challenge of developing targeting molecules and labeling them with SERS-tags. Label-free SERS has therefore been particularly successful where target-binding entities have not yet been established, and where spectral information can inform of changes in molecular structure. There is of great value when the compositional diversity of molecules that is encoded in the unique SERS signatures is interrogated instead of limiting to specific molecules as a target. Indeed, label-free SERS has enabled detection of biomolecules[7–12], drug monitoring[13–15], studies of molecular structures of biomolecules[16–18], identifying biological species[19–22], diagnosing diseases[23–25], through to monitoring biological processes at the cellular level[26–28]. Despite the promises that arise from the sensitivity of SERS to molecular orientation and separation from the plasmonic surface, there are limitations due to the inherent chemical and structural complexity of biomolecules that yield overlapping spectra. This is often viewed as an insurmountable challenge necessitating methods to specifically bind target analytes that may otherwise present low Raman scattering cross sections[29] and weak affinities to the plasmonic surfaces of SERS sensors or may be outcompeted by binding of abundant components of biological systems[30,31].

Self-assembled monolayers (SAMs) have been explored in several different ways to introduce higher selectivity towards specific analytes at the plasmonic surface of SERS sensors. For instance, zwitterionic SAMs can resist nonspecific fouling of proteins in complex media and minimize their contribution to SERS spectra[31,32]. Combinations of SAMs have been tailored to improve selectivity towards certain small molecules with low Raman scattering section such as glucose[33–37] and 3,4-methylenedioxymethamphetamine (MDMA)[38]. These prior studies have focused on the effective optimization of the SAM composition to improve selectivity without a specific receptor[33,34,38], or to minimize the influence of off-target biological system components to enhance the analyte signal[31,32]. Such approaches aim to reduce the complexity of SERS fingerprints. However, there is significant potential in general methods that can capitalize upon the rich compositional information present within the overlapping spectra rather than attempting to minimize it. Artificial-nose approaches represent a promising strategy towards embracing the compositional diversity when interrogating biological samples[39–41]. In such approaches, low-specificity physicochemical interactions at arrays of different sensing receptors are used, each yielding physical signals that can be recorded and combined to generate a patterned output. The effectiveness of this addition of data dimensionality from an array of output channels is assisted by chemometric data analysis to build classifiers towards hypothesis-free sample identification. These approaches, however, frequently monitor one-dimensional outputs such as fluorescence intensity[42,43], electrical response[44] or mass change[45] per detection receptor. Label-free SERS in contrast offers the potential for two-dimensional readouts per receptor drastically amplifying the amount of chemical and structural information obtainable for a small arrayed sensor system.

We demonstrate that it is possible to substantially enhance the accuracy of biological sample identification without target-specific binding receptors by increasing data dimensionality through multiple SAM-functionalized surfaces while embracing the spectral complexity of the label-free SERS datasets. The approach is employed as an arrayed sensing platform, which we term "Functionalized Array for Surface-Enhanced Raman Spectroscopy (FASERS)", consisting of eight different SAMs formed on plasmonic Au-nanopillars. A series of un-targeted, mildly selective SAMs have been employed in our system to promote diverse ranges of physicochemical interactions with different sample constituents rather than improving the detection of pre-selected target molecules or minimizing the off-target biological system components. Through selective SERS enhancement of molecules in close proximity to each surface, we sought to present the diversified SERS signatures detectable in complex liquids. In this approach, the compositional diversity of biologically derived liquids, which has previously limited conventional label-free SERS biosensors, is leveraged by introducing another dimensionality to the obtained datasets. We illustrate these varied interactions by assessing the binding of four small molecules to the SAMs of differing molecular characteristics and explore the manifold range of interactions at play using molecular dynamics simulations. The SAM-dependent multi-dimensional spectral datasets can identify and discriminate complex biological samples with higher accuracy compared with conventional label-free SERS. The merit of this approach is shown for two cell lysates from companion cell lines that comprise malignant and normal human cell lines from the same tissue of a patient, whereby combining the spectral signatures for multiple SAMs, the classification accuracy of label-free SERS can be improved in a facile manner. Our approach highlights that by exploiting label-free SERS and its ability to interrogate biomolecules as a function of SERS sensor surface functionality, a powerful artificial-nose empowered strategy can be envisaged.

## Results

**Design and fabrication of FASERS.** The arrayed artificial-nose sensing platform FASERS comprises eight different SAM-functionalized Au-nanopillar substrates and one bare unmodified substrate. To promote a diverse range of physicochemical interactions between analytes and the SAMs within SERS-active regions, we utilized SAM-forming molecules of varied molecular characteristics which presented four functional end-groups with two different lengths of the carbon chains (Table 1). These surface

---

**Table 1 Self-assembled monolayer (SAM) forming molecules used in FASERS fabrication.**

| Functional group | 3-Carbon chain | 11-Carbon chain |
|---|---|---|
| Alkyl | 1-Propanethiol ($3CH_3$) | 1-Undecanethiol ($11CH_3$) |
| Hydroxyl | 3-Mercapto-1-propanol ($3OH$) | 11-Mercapto-1-undecanol ($11OH$) |
| Carboxyl | 3-Mercaptopropionic acid ($3COOH$) | 11-Mercaptoundecanoic acid ($11COOH$) |
| Amine | 3-Amino-1-propanethiol ($3NH_2$) | 11-Amino-1-undecanethiol ($11NH_2$) |

chemistries with differing charge and hydrophobicity were designed to promote heterogeneous interactions ranging from electrostatic and hydrogen bonding interactions to van der Waals interactions. The change of carbon lengths not only serves to provide different thicknesses of hydrophobic domains, but also to give a graded potential for SERS signal enhancement by determining the distance of the end-group from the gold surface with respect to the first decay length of SERS signals ($d_{1/2}$, ~7 Å)—the characteristic distance at which the signal reduces by half[46].

The chemical structures of the eight SAMs are designed to influence the composition, concentration, distance and/or orientation of molecules residing at equilibrium within SERS-active regions. The differences in the types of molecules that are present but also the molecular behaviour in or proximal to hotspots can subsequently be profiled by label-free SERS. Recording a label-free SERS spectrum of a solution from each SAM-functionalized receptor yields nine spectral outputs that can constitute a higher dimensionality signature. The resulting signatures can then be utilized to identify and classify the samples with higher specificity and enhanced accuracy through multivariate analysis techniques, such as principal component analysis (PCA) and linear discriminant analysis (LDA) as highlighted in Fig. 1.

We fabricated SERS-active gold film-coated nanopillar substrates (Au-nanopillars) via colloidal lithography and plasma etching as described in Fig. 2a and Supplementary Fig. 1a, b. We demonstrated SERS activity of the fabricated Au-nanopillars with 4-aminothiophenol (ATP) that was chemisorbed onto the surface via Au–S bonds. The obtained SERS spectra of ATP were in accordance with previously reported SERS spectra of ATP adsorbed onto gold nanorings[47] and gold nanorods[48]. The two prominent characteristic peaks of ATP located at 1078 and 1586 cm$^{-1}$, correspond to C–S stretching and C–C stretching, respectively (Fig. 2b, inset). The peak intensities increased with the increasing ATP concentration, up to a saturation point that likely correlated with complete coverage of the SERS-active surface (Fig. 2b, Supplementary Fig. 1c). To establish the spatial

reproducibility of the SERS enhancement, we performed SERS mapping on Au-nanopillars substrates using the prominent peak area (1057.1–1091.7 cm$^{-1}$), centred at 1074.7 cm$^{-1}$, of chemisorbed 1 μM 4-mercaptobenzoic acid (4-MBA) (Supplementary Fig. 2). Significant SERS enhancement of 4-MBA signals across the entire scanned area was observed compared with those of a flat gold film-coated surface (Au–Si). Importantly, we observed little to no significant signal variation between the different spatial locations, indicating the robust spatial reproducibility of the SERS substrate. We recorded background SERS signals for one non-functionalized (bare) and eight SAM-functionalized Au-nanopillar arrays in phosphate buffered saline (PBS) (Table 1, Fig. 2c), where we observed reproducible spectra as highlighted by the minimal standard deviation across multiple substrate replicates. We did not detect any significant SERS signatures for the bare Au-nanopillars while SERS signatures with varying degrees of intensity were detected for the functionalized Au-nanopillars. In particular, relatively strong signatures were obtained from the 1-alkanethiols, consistent with reported literature[49–52], which is attributed to the lowest symmetry along the vibration axis and large contribution from stretching of *trans*-conformers. The four ligands with 11-carbon alkyl chains showed comparable SERS signatures with a predominant peak at about 1100–1200 cm$^{-1}$, which corresponds to symmetric the C–C stretching mode being the largest contribution from the tensor component along the axis of the vibration[49,51]. Detailed tentative assignments of the peaks are further discussed in Supplementary Table 1.

We prepared SAMs by ethanolic immersion of gold film-coated silicon wafers (Au–Si) with pH adjustment where necessary (see the Methods section) and verified the efficacy of the protocol by water contact angle measurements, atomic force microscopy (AFM) and X-ray photoelectron spectroscopy (XPS). The role of the various SAM headgroups and chain lengths in determining the interfacial properties of the gold surface manifested as a clear variation of contact angles between 19° and 105° (Fig. 2d, Supplementary Fig. 3b–d) compared with 52° for the cleaned gold surface, consistent with previously reported

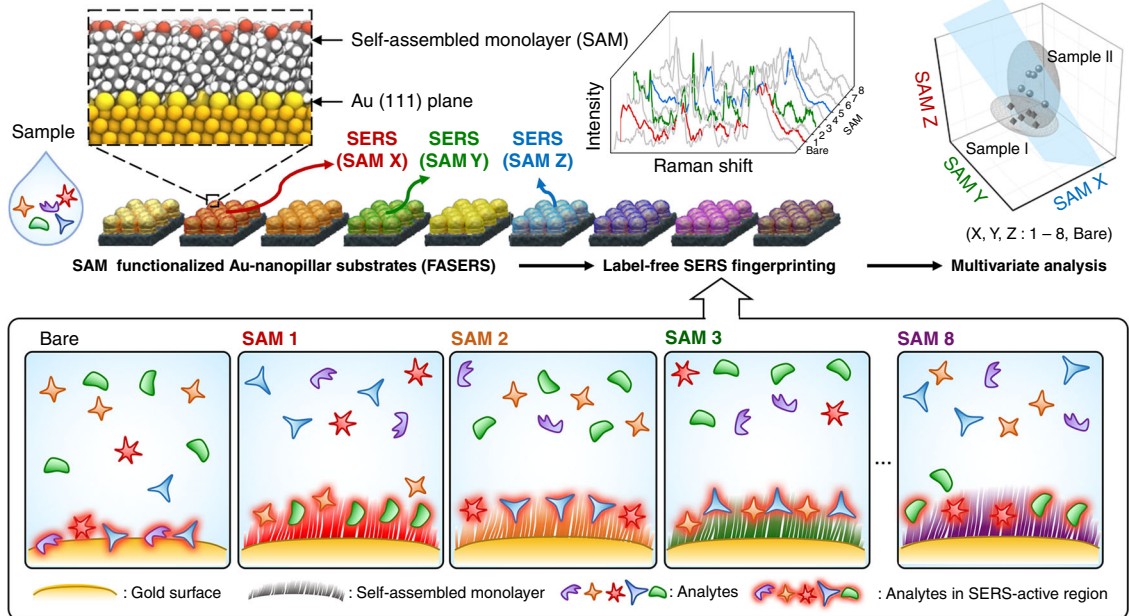

**Fig. 1 Schematic illustration of artificial-nose-empowered surface-enhanced Raman spectroscopy.** 'Functionalized Array for Surface-Enhanced Raman Spectroscopy (FASERS)' represents an array of plasmonic surfaces for label-free SERS presenting different self-assembled monolayers. A range of molecular interactions takes place within complex biological media at each unit sensor where mildly selective SERS enhancement of the constituents gives multiplexed spectral datasets. The increased data dimensionality obtained enables facile identification of closely related samples.

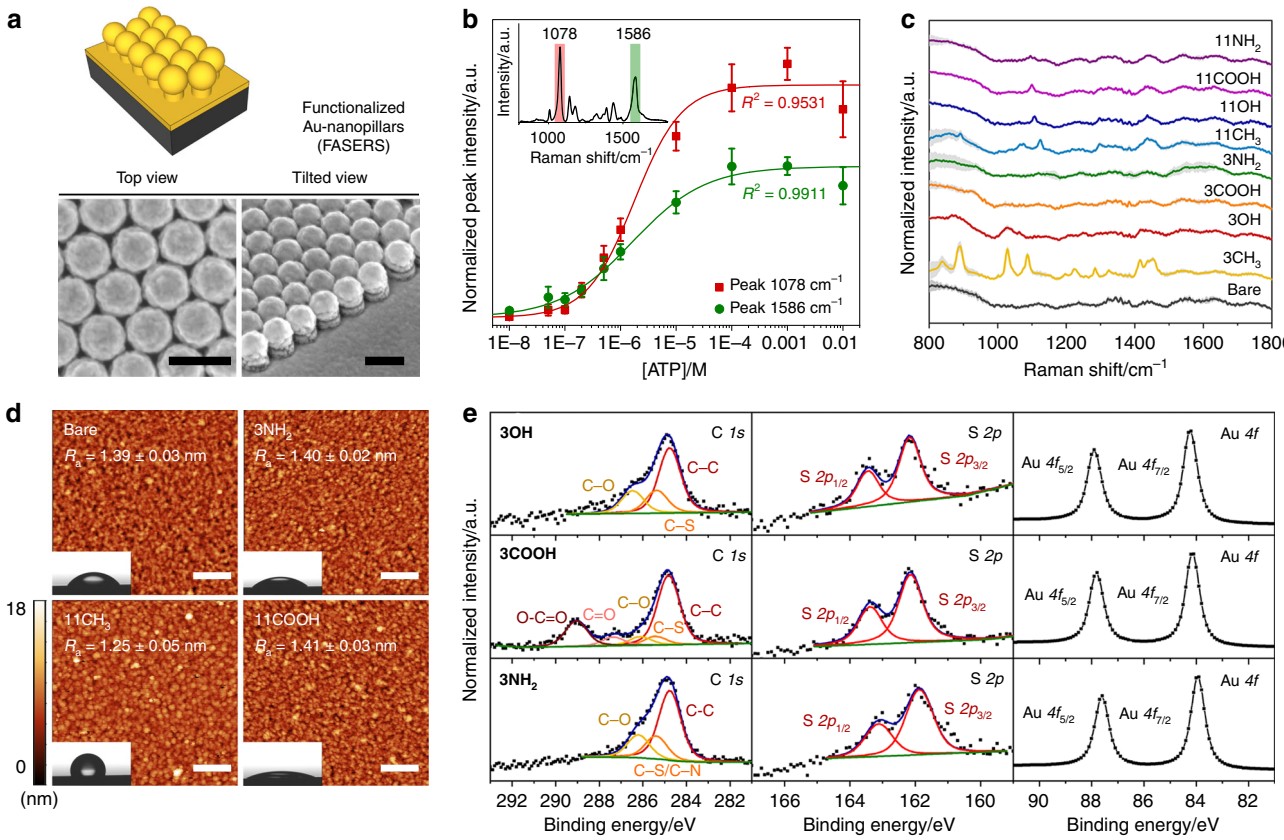

**Fig. 2 Fabrication and characterization of FASERS substrates. a** Schematic illustration and representative SEM images of gold film-coated polystyrene beads (PS)–Si₃N₄ (Au-nanopillars) SERS-active substrates that are used in FASERS. SEM images were obtained from top view and angled view from 45° tilted stage without extra metal coating. Scale bar, 400 nm. **b** Normalized SERS intensities of two prominent peaks of 4-aminothiophenol (ATP) with varying concentration on non-functionalized Au-nanopillars. Data represent mean ± 1 s.d. from nine obtained spectra ($N = 3$, $n = 3$ spectra) and fitted with a sigmoidal curve. Inset graph shows the mean of nine spectra from a 1 mM ATP solution. **c** SERS spectra of functionalized Au-nanopillars in PBS (10 mM, pH 7.4) with various SAM-forming molecules: non-functionalized (bare), 1-propanethiol (3CH₃), 3-mercapto-1-propanol (3OH), 3-mercaptopropionic acid (3COOH), 3-amino-1-propanethiol (3NH₂), 1-undecanethiol (11CH₃), 11-mercapto-1-undecanol (11OH), 11-mercaptoundecanoic acid (11COOH), 11-amino-1-undecanethiol (11NH₂). Solid lines and grey shaded area represent mean and ±1 s.d. of nine obtained spectra ($N = 3$, $n = 3$ spectra). **d** Representative AFM images of several of the functionalized gold film-coated Si wafers (Au–Si) with various SAM-forming molecules (from top-left to bottom-right): Bare, 3NH₂, 11CH₃, 11COOH. Scale bar, 400 nm. Average of the mean roughness ($R_a$) of each surface was noted in the image with ±1 s.d. ($n = 3$ scans). Inset images indicate the range of water contact angles observed at each surface. **e** Representative high-resolution XPS spectra of C *1s*, S *2p* and Au *4f* for 3OH (top), 3COOH (middle) and 3NH₂ (bottom) SAM-functionalized Au–Si.

values for SAMs[53–55]. To ensure that the changes in contact angles did not originate from changes in the surface roughness and topography of the gold film, we performed AFM on Au–Si (Fig. 2d, Supplementary Fig. 3a). AFM revealed only minor changes in the surface roughness ($R_a$), indicating that the SAM functionalization conditions could dictate the surface chemistry without causing reshaping of the surface. To further characterize the SAM modified gold surfaces, we performed XPS on Au–Si to assess the chemical structures of surface-bound molecules. High-resolution XPS scans of C *1s*, S *2p* and Au *4f* regions were compared before and after the SAM functionalization (Fig. 2e and Supplementary Fig. 4). Deconvolution of the C *1s* and S *2p* peaks into the corresponding chemical states confirmed that the SAMs chemisorbed via thiol–Au bonds without significant contaminants.

**Modulation of physicochemical SERS signatures using functionalized plasmonic surfaces.** SERS enhancement of a molecular vibrational mode is known to be modulated in a distance- and orientation-dependent manner where Raman modes in close proximity to the SERS-active surface and with an orientation perpendicular to the surface are preferentially enhanced. In order to demonstrate SAM-dependent SERS output signatures from each receptor, we measured four model molecular solutions of differing charge, size and hydrophobicity by FASERS (Fig. 3): *p*-phenylenediamine (*p*-PDA), 4-aminophenylacetic acid (4-APA), Rhodamine 6G (R6G) and folic acid (FA). To obtain insight into the underlying interactions at play, we investigated the two closely matched small molecules with opposite electrostatic potential, 4-APA and *p*-PDA at each SAM interface with all-atom molecular dynamics (MD) simulations (Fig. 4, Supplementary Table 4, Supplementary Figs. 5 and 6). This enabled the relative analyte orientation, distance to the gold surface, and contact lifetime with the SAM/gold surface to be obtained (Fig. 4, Supplementary Fig. 7, Supplementary Tables 5 and 6) within defined SERS-active regions (<ca. 0.6 nm from the end-groups of SAMs), resultant from the simulated intermolecular interactions.

The SERS spectral intensities and profile changes of the four test molecules can be qualitatively understood by assessing the absolute and relative intensities of the prominent peaks within the spectra (Fig. 3b–i, Supplementary Tables 2 and 3). Each peak corresponds to a unique molecular vibrational mode and each molecule is found to clearly exhibit a number of spectral features

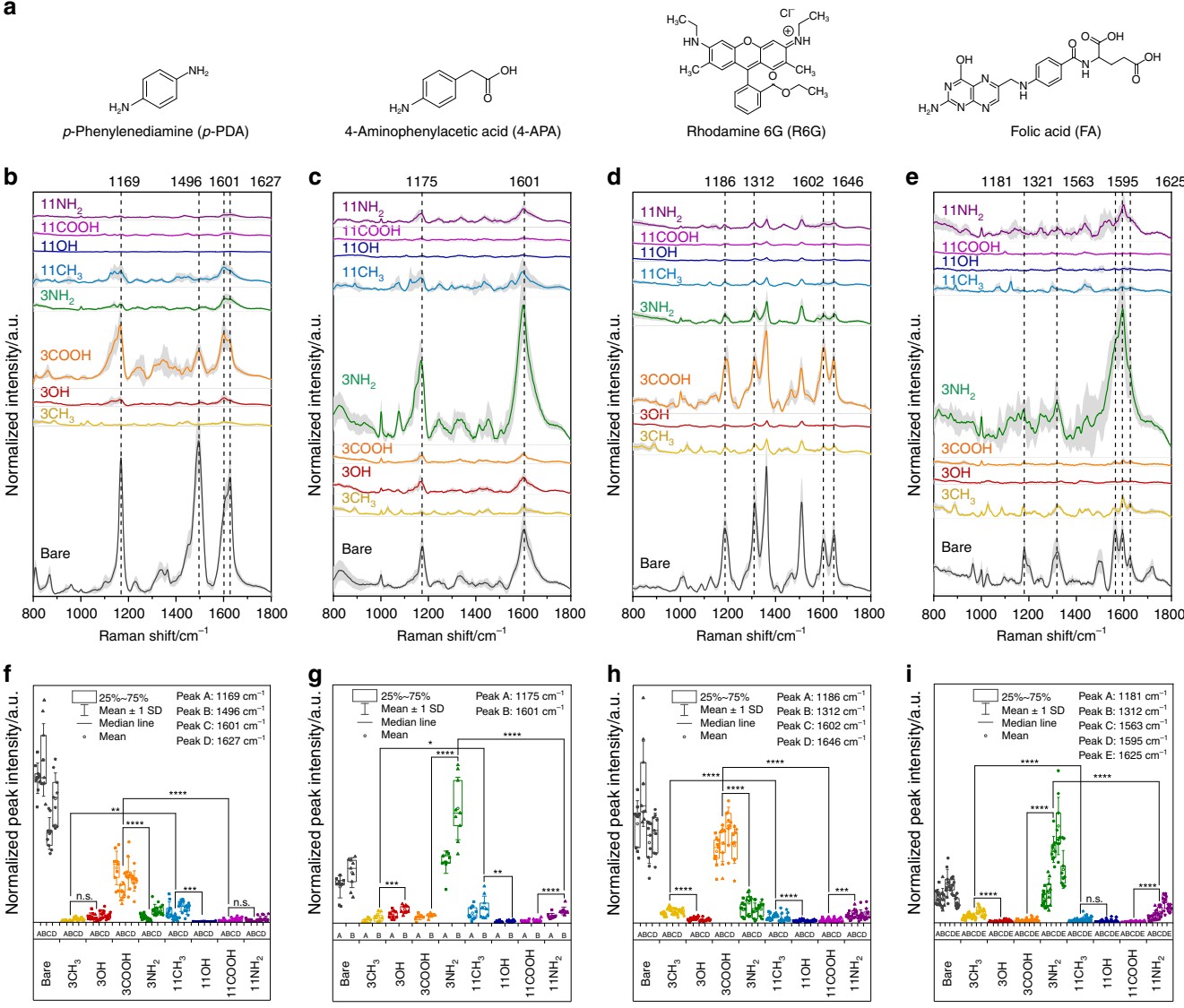

**Fig. 3 Physicochemical SERS fingerprints of molecular analytes using FASERS. a** Chemical structures of the four model analyte molecules. **b–e** Series of SERS spectra obtained from functionalized Au-nanopillars with various SAM-forming molecules: non-functionalized (bare), 1-propanethiol (3CH$_3$), 3-mercapto-1-propanol (3OH), 3-mercaptopropionic acid (3COOH), 3-amino-1-propanethiol (3NH$_2$), 1-undecanethiol (11CH$_3$), 11-mercapto-1-undecanol (11OH), 11-mercaptoundecanoic acid (11COOH), 11-amino-1-undecanethiol (11NH$_2$). **b** 500 μM *p*-phenylenediamine (*p*-PDA) in phosphate buffer (10 mM, pH 5.0), **c** 500 μM 4-aminophenylacetic acid (4-APA) in phosphate buffer (10 mM, pH 7.5), **d** 100 μM Rhodamine 6 g (R6G) in phosphate buffer (10 mM, pH 5.0), and **e** 500 μM folic acid (FA) in phosphate buffer (10 mM, pH 7.5). Solid lines and grey shaded areas represent mean and ± 1 s.d. of nine obtained spectra (N = 3, n = 3 spectra). **f–i** Peak analysis on the prominent peaks of the solutions; **f** *p*-PDA, **g** 4-APA, **h** R6G and **i** FA. Data represent the peak intensities determined from the nine spectra (N = 3, n = 3 spectra). ****$p < 0.0001$, ***$p < 0.001$, **$p < 0.01$ and *$p < 0.05$ based on one-way ANOVA and Tukey's honest significance test.

and intensities that vary as a function of SAM composition across repeated measurements (Fig. 3b–e). The variation in molecular orientation and proximity to the gold surface that underpins these responses can also be seen in the diverse equilibrium configurations of 4-APA and *p*-PDA at each SAM obtained from our MD simulations (Fig. 4a). One of the most prominent effects observed in both SERS measurements and the MD simulations is the impact of electrostatic interactions. Indeed, both FA and 4-APA yielded a stronger SERS response in the presence of the 3NH$_2$-SAM versus the bare-gold surface, despite the presence of a SAM covering the Au surface. On the other hand, *p*-PDA and R6G showed the most pronounced response at the 3COOH-SAM following the bare Au surface. These differences emphasize the potential for SAM-dependent interactions that can promote

different molecular orientations or binding affinities near the metal surface promoting different SERS spectral responses. The simulations of 4-APA and *p*-PDA likewise revealed that analyte–SAM interactions are predominantly electrostatically driven, with the most and least persistent contacts forming between moieties carrying the opposite and same charges, respectively. This is further highlighted in Fig. 4b, c where well-defined analyte orientational distributions with high probability density peaks (percentage populations) were observed. These strong analyte–SAM interactions likely contribute to the distinct changes in the spectral profiles of the molecules on SAMs bearing opposite charges as highlighted by changes in the relative peak intensity ratios (Fig. 3). This is consistent with the previous observations that the orientation angle likely plays an important

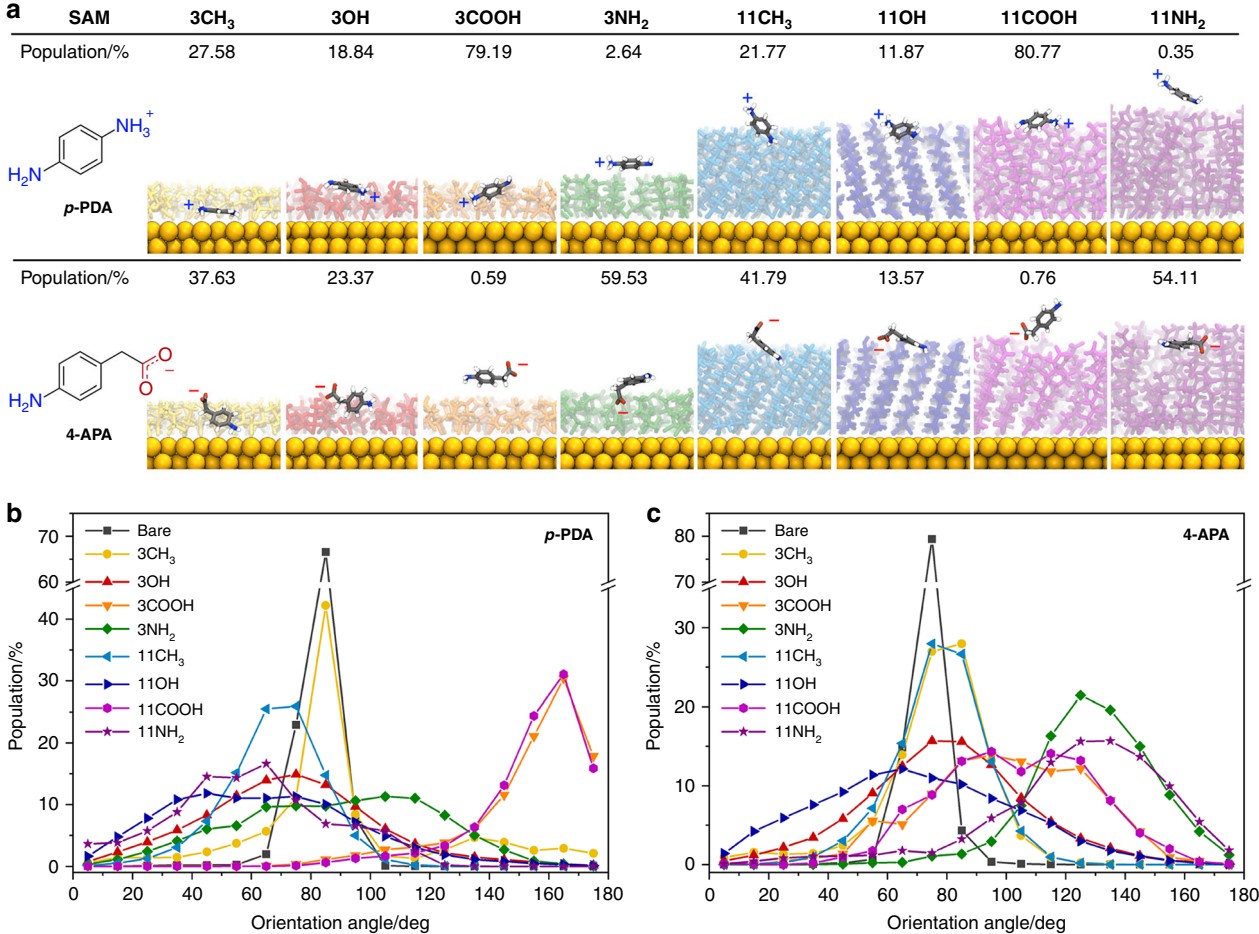

**Fig. 4 Molecular dynamics simulation of analytes on SAM-functionalized gold surfaces. a** Snapshots of two model analytes, *p*-phenylenediamine (*p*-PDA) and 4-aminophenylacetic acid (4-APA), where the analyte–gold surface separation reaches a minimum. Analyte center-of-mass is used to measure the distance to the closest Au surface atom; % populations when analytes proximal (<0.6 nm) to a SAM/Au are calculated from MD generated equilibrium ensemble for each system: non-functionalized (bare), 1-propanethiol (3CH$_3$), 3-mercapto-1-propanol (3OH), 3-mercaptopropionic acid (3COOH), 3-amino-1-propanethiol (3NH$_2$), 1-undecanethiol (11CH$_3$), 11-mercapto-1-undecanol (11OH), 11-mercaptoundecanoic acid (11COOH), 11-amino-1-undecanethiol (11NH$_2$). Analyte orientation angles of **b** *p*-PDA and **c** 4-APA relative to the Au surface showing the direction of each analyte's functional groups when proximal (<0.6 nm) to a SAM/Au. Angles < 90° indicate that the NH$_2$ group of the analyte is pointing towards the SAM/Au surface, whereas angles > 90° specify that the charged groups (NH$_3^+$/COO$^-$) are facing the SAM/Au. SAM and analyte protonation states are modeled based on the cognate experimental buffer conditions, i.e. *p*-PDA (pH 5.0) and 4-APA (pH 7.5).

role in the excitability of Raman active bonds in the molecules as illustrated here by the appearance of the characteristic shoulder band (~1140–1150 cm$^{-1}$) of *p*-PDA and 4-APA.

While the electrostatic forces may play a dominant role with respect to signal intensity, we observed considerable differences between the SERS responses of the methyl- and hydroxyl-terminated SAM with respect to spectral profiles and the extent to which different Raman modes are excited. These SERS signals were less prominent than those of the amine or carboxyl-bearing SAMs indicating a more subtle role of these additional weaker interactions in defining a SERS response of the system. For example, the average orientation of both *p*-PDA and 4-APA relative to the gold surface remained similar owing to hydrophobic interactions between the benzene ring and the aliphatic chains. Hydroxyl-terminated SAMs, on the other hand, formed hydrogen bonds with the unprotonated amine moiety of the analytes. Overall, the simulations of the 11-carbon chains showed that 4-APA and *p*-PDA were typically present at an average distance of ~2 nm from the gold surface, which is reflected in their lower SERS intensities compared with the 3-carbon chains. In the case of the thicker hydrophobic 11CH$_3$

SAM, *p*-PDA oriented at a well-defined angle with partial embedding of its amine groups into the SAM interface. Such partitioning leads to a perpendicular orientation angle that is expected to promote a more effective SERS enhancement. The MD simulations indicated that the SAM structure and dynamics could influence the mechanism of 4-APA and *p*-PDA interactions with the SAM or the small molecule's approach to the Au surface. For example, the CH$_3$-terminated SAMs enabled a closer contact of the molecules with the nanoparticle surface (Fig. 4a, Supplementary Fig. 7, Supplementary Tables 5 and 6) due to hydrophobic clustering/aggregation of ligands that exposed the bare nanoparticle surface in the short chains systems and defect sites/grooves in the long chain systems. In addition, MD simulations of the 11NH$_2$–SAM suggested a degree of amine-Au interaction (Fig. 4a) leading to hydrophobic pockets better able to sequester 4-APA. These observations highlight that the specific SAMs' structural and dynamic properties could play a key role in their efficacy in binding certain molecules.

The MD simulations shed light on some of the mechanisms through which the FASERS strategy establishes physicochemical fingerprints, whereby SERS enhancement stems from the

combined effects of strength and persistence of the intermolecular interactions, surface distance and orientation of analytes. In such systems, each of the nine receptors plays a role in generating differential binding profiles of molecules in the sample by reflecting the preferred orientation and distance of analytes upon equilibration of physical forces in the system. The test molecule experiments are much simpler than those expected in complex biological media but establish some of the interaction mechanisms that might be anticipated. Although the countless number of chemically distinct molecules in such systems will create significantly more complex signatures on each SAM due to diverse analyte–SAM binding distances and characteristic orientations, the interaction profiles on each of the eight SAMs are still likely to be unique.

**High-dimensionality fingerprinting of biological samples by FASERS.** Unlike those of single component molecule solutions, the SERS spectra of biological samples are expected to be more complex resulting from the overlapped signatures of a myriad of biomolecules, spanning proteins, lipids, nucleic acids, glycogens and metabolites. As a demonstration of a compositionally complex, biologically derived system, but in controlled, purified media, we employed FASERS for investigation of extracellular vesicles (EVs) isolated from MDA-MB-231 human breast cancer cells (Supplementary Fig. 8). EVs used in the analysis were purified by size exclusion chromatography to remove soluble protein and other components present in the secretome, to ensure the label-free SERS signatures from the solution could be attributed to the variation of EV composition and orientation near the surface rather than artefacts from other impurities. Peak positions, shapes and intensities within the EV fingerprints corresponding to proteins, lipids, and nucleic acids varied significantly depending on SAM functionality. The implication of this is that the EVs were interacting at the gold or SAM surfaces enabling the SERS-active regions to survey the EV composition near the surface. Alternatively, it could reflect a degree of dissociation and release of their contents during interaction with different SAMs. Either way, the variation observed demonstrates the merit of multiple SAM output channels in a controlled complex sample.

Having established the feasibility of diverse signal generation using FASERS in a complex yet controlled EV biological system, we employed an artificial-nose empowered approach to validate how obtaining spectral data as a function of sensor surface chemical functionality can improve discrimination of complex biological samples. To achieve this, we have employed FASERS to interrogate lysed Hs578T breast carcinoma cells and Hs578Bst normal fibroblast-like cells from the same individual—human companion cell lines that have been intensively used in the study of breast cancers since 1977[56,57]. Cell lysates, prepared from both cell lines using identical procedures, were dropped onto the FASERS substrates and their spectra collected. The SERS spectra obtained are shown in Fig. 5a–b. The characteristic SERS bands originate from the vibrational modes in proteins (i.e. C–C–N*, C–N, amide, C–C, C=C vibration), carbon chain vibrations in lipids (C–H deformation, C–H$_2$ aliphatic twist), as well as various in-/out-of-plane vibrational modes within the nucleobases (i.e. adenine (A), cytosine (C), guanine (G), thymine (T), uracil (U)) and O–P–O stretching in nucleic acids[58–62]. The non-functionalized Au-nanopillars exhibited limited variation in their SERS fingerprints between the two cell lysates, and discrimination based on the band positions and peak intensities is not straightforward. This is attributed to the inherent chemical/structural complexity and low Raman scattering cross-section of biomolecules, thus minimal differences in concentration and composition are not readily detectable[5,29,63]. Moreover, it is

noteworthy that the SERS spectra on the bare-gold surfaces were largely dominated by protein bands that were adsorbed on gold surfaces. Although protein is one of the key elements in the biological samples, the adsorption of the most abundant proteins often leads to saturation of most of the SERS hotspots and can hinder other small biomolecules from approaching[32].

We observed a number of distinct differences in the SERS signatures for each of the functionalized Au-nanopillars between the two cell lysates compared with the non-functionalized Au-nanopillars. These differences included enhanced and/or suppressed peak intensities as well as apparent position shifting and spectral shape changes. This indicates that the approach of modulating the fraction of components within a complex sample that interact via mildly-selective physical interaction at the SAMs enabled the generation of eight additional spectral profiles of the complex samples. In order to quantitatively establish the classification potential of using FASERS, the SERS data obtained from the two cell lysates was analysed using an artificial-nose-inspired approach based on statistical multivariate analysis (Fig. 5c–h, Supplementary Figs. 9–11, Supplementary Table 7). To investigate the effect of each functionalization, the SERS data from the non-functionalized and eight SAM-functionalized substrates were initially analyzed individually by principal component analysis (PCA). The data were pre-processed by mean centering and the first principal component (PC1) was calculated for each substrate, resulting in a total of nine PC1s used for further analyses (Supplementary Table 7). Comparing the PC loadings of the nine PCs obtained for each functionalization and the tentative assignment of the SERS characteristic bands (Fig. 5a, b), it is clear that the locations of large variability in the loadings are consistent with the bands (Fig. 5c) attributable to endogenous biomolecule variations (DNA, proteins and lipids). Each SAM contributed unique Raman signatures of the different cell lysates, which demonstrates the capability of selectively screening biomolecular constituents within the cell lysates. By overlapping the PC loadings with the inherent SERS signatures of the SAMs alone, we further confirmed that the variations used for discrimination did not originate from the substrates but were inherently present within the samples (Fig. 5c).

Figure 5f shows a scatter plot of the first two PCs for the non-cancerous and cancerous cell lysates on the non-functionalized Au-nanopillars. The corresponding accuracy of the discrimination was 75% and a significant overlap of the clusters from the two cell lysates was observed. The implication is that the spectral differences between the two samples were not sufficient to fully discriminate using the non-functionalized substrates alone. Inspired by an artificial-nose-based approach, we performed a non-conventional PCA–linear discriminant analysis (LDA) where the PC1s from multiple SAMs were cross-combined for analysis. In this approach, instead of increasing dimensionality by employing several PCs (e.g. PC1, PC2, PC3, PC4 etc.) from the same dataset as in conventional PCA–LDA, we have utilized the first PC (PC1) from multiple SAMs and gradually increased the number of SAMs included in the analysis to show the merit of amplifying the dimensionality of information. Figure 5g shows representative two-dimensional PCA scatter plots for PC1s from two different SAMs. Significantly, the resulting separation line derived from LDA could discriminate the two cell lysates with 100% accuracy (Fig. 5g), which was initially not achievable using non-functionalized substrates (Fig. 5f). This indicates that modulation of SERS signatures and combination of resulting compositional fingerprints can yield improved accuracy. MD simulations (Fig. 4) highlighted the importance of electrostatic and hydrophobic interactions in defining the molecular interaction at the SAMs and indeed, the combinations of the amine-, carboxyl- and alkyl-terminating SAMs show the highest accuracy

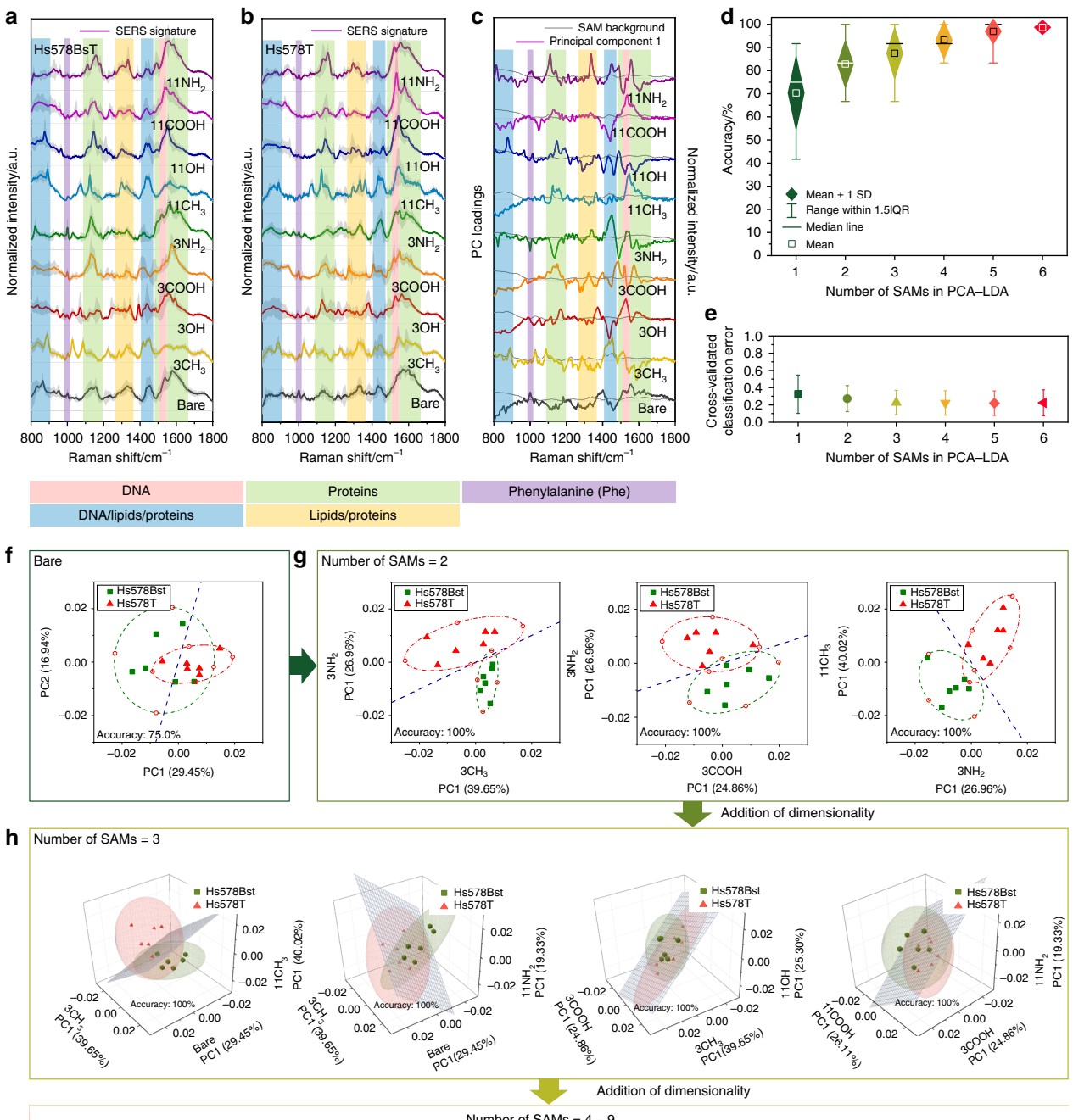

**Fig. 5 Artificial-nose-empowered statistical multivariate analysis of model biological systems. a**, **b** Series of SERS spectra from cell lysates obtained from functionalized Au-nanopillars with various SAM-forming molecules: non-functionalized (bare), 1-propanethiol (3CH$_3$), 3-mercapto-1-propanol (3OH), 3-mercaptopropionic acid (3COOH), 3-amino-1-propanethiol (3NH$_2$), 1-undecanethiol (11CH$_3$), 11-mercapto-1-undecanol (11OH), 11-mercaptoundecanoic acid (11COOH), 11-amino-1-undecanethiol (11NH$_2$). **a** Hs578Bst normal fibroblast-like cells, and **b** Hs578T breast carcinoma cells. Solid lines and grey shaded area represent mean and ± 1 s.d. of six obtained spectra ($N = 3$, $n = 2$ spectra). Coloured shaded bands refer to tentative assignments established in the literature[58–62]. **c**–**h** Principal component analysis (PCA)–linear discriminant analysis (LDA) for the two cell lysates: **c** First principal component (PC1) loadings of SERS spectra from each functionalization. Overlapped grey lines indicate the background SERS signatures of the substrates in PBS (10 mM, pH 7.4). **d** Mean accuracy for discrimination of cancerous cell lysates (Hs578T) from non-cancerous cell lysates (Hs578Bst) using PCA–LDA models, calculated at each dimensionality from all possible cross-combinations of the nine PC1s. **e** Evaluation of predictive performance of the PCA–LDA models in (**d**) using leave-one-out cross-validation (LOOCV). CV-classification errors represent the misclassified fraction of the observations for each LOOCV model. Data represent mean ± 1 s.d. of all the models at each dimensionality. **f** Scatter plot of the first principal component (PC1) versus the second principal component (PC2) from non-functionalized Au-nanopillars. **g** Representative two-dimensional PCA scatter plots for the two cell lysates using PC1s of two different SAM functionalization. Blue dotted line was derived by LDA as a classification algorithm to separate the two groups. Red and green dotted boundaries represent confidence intervals of the ± 1 s.d. of each group. Inset is the calculated accuracy in cancerous cell lysates (Hs578T) discrimination for each model. **h** Representative three-dimensional PCA scatter plots with each axis corresponding to the PC1s of three different SAM-functionalized Au-nanopillars. Blue planes depict classification derived from LDA algorithm separating the two groups. Red and green ellipsoids represent ± 1 s.d. of each group.

(Fig. 5g, Supplementary Fig. 9). We introduced a further data dimensionality in the PCA by cross-combining three different PC1s (Fig. 5h) from different functionalized surfaces to generate three-dimensional PCA scatter plots. Notably, the LDA-derived planes could discriminate the two cell lysates with 100% accuracy in 20 PCA three-dimensional PCA scatter plots among all of the 84 possible combinations (Fig. 5h, Supplementary Fig. 10). Figure 5h shows representative three-dimensional PCA scatter plots of 100% accuracy that cannot be achieved when using any of the two axes in a 2D configuration. This improvement implies that increasing data dimensionality can improve the classification power of the system.

Although impossible to plot, PCA–LDA were performed for higher dimensionalities up to a maximum of nine-dimensions in which the nine PC1 from different functionalized substrates were simultaneously utilized. In order to evaluate the performance at each dimensionality, all of the possible cross-combinations of the nine PC1 at each number of SAMs were considered (Fig. 5d–e, Supplementary Fig. 11). The mean accuracy in discrimination of two cell lysates using the calculated PCA–LDA models (Fig. 5d, Supplementary Fig. 11a) showed an apparent increase when adding more axes of different functionalization. Indeed, when six functionalities were used, 100% accuracy was achieved in 71 six-dimensional PCA among all of the 84 possible combinations. To estimate the quality of classification, all PCA–LDA models at each dimensionality were validated by performing leave-one-out cross-validation (LOOCV) (Fig. 5e). During LOOCV, a single sample (i.e. observation) is retained as a validation set while the remaining samples are used as training set. It is followed by a classification test using the left-out sample, and this training-test process is iterated for all the samples available in the set. The fraction of misclassified observations out of the total number of the observations was calculated, referred to as cross-validated (CV) classification error, which is commonly used for assessing predictive performance of the PCA–LDA models quantitatively. Figure 5e shows the mean CV-classification error calculated for all PCA–LDA models at each dimensionality. Importantly, increasing the number of included functionalities yielded a reduction in the CV-classification error, confirming the improvement of predictive capability of the classifier using modified PCA–LDA of FASERS. The impact of sample size and higher dimensionality analysis are discussed further in Supplementary Fig. 11.

This implementation of an artificial-nose empowered SERS approach to generate and analyze high-dimensionality data by increasing the number of spectral outputs from a single sample represents a proof-of-concept artificial-nose based sensing methodology. Different sampling constituents can be screened per substrate with enhanced selectivity without using any targeting receptor and as a result of the multiple SAM-dependent spectra, we achieved the discrimination of closely related biological samples with enhanced accuracy, compared with when only the more traditional bare substrates were used. Extending this initial demonstration of the FASERS platform to other biologically relevant systems is the next step in evaluating the potential for sample characterization and discrimination. Particularly, an in-depth clinical study with patient-derived biological fluids such as blood plasma represents an important follow-up research direction towards a hypothesis-free diagnostics platform. We propose this could be realized by careful selection of SAMs and automated methods to collect, statistically analyze, and interpret these datasets such as artificial-intelligence-based feature selection. When expanding to clinical-derived systems with larger sample sizes, this would enable optimal model parsimony to reduce aberrant noise and other spurious signatures that do not contribute to sample discrimination. The system represents a hypothesis-free approach that could be exploited towards accurately discriminating the differences between distinct populations but also to identify the common similarities among closely related populations. This offers a platform technology to characterize, cluster and classify biological samples, which in the context of diseases, such as cancer, opens up numerous possibilities for a simple yet powerful label-free diagnostic platform.

## Discussion

We have developed an artificial-nose inspired label-free arrayed SERS sensor by implementing the low-specificity physicochemical selectivity of SAMs, without any target-specific binding entity, to obtain multiple compositional fingerprints. As an artificial-nose-like sensing approach, our comprehensive interpretation of the increased sample data improved the discrimination of complex solutions by selectively screening and emphasizing the approach of different components of the samples to the sensors depending on the molecular characteristics of each SAM. The four model molecular solutions as well as two model biological samples successfully demonstrated the spectral diversity possible by employing the FASERS system. Through the use of a modified multivariate analysis employing combinations of the first principle components of differing numbers of surface functionalities, we have shown that increasing the number of surface functionalities used in the sample interrogation enhances accuracy in the classification of highly heterogeneous and complex biological systems such as cell lysates. This system highlights a strategy to improve molecular selectivity and potentially the reproducibility of label-free SERS by increasing the output-data dimensionality where label-free SERS represents a powerful additional tool for artificial-nose sensing. Given the versatility and broad applicability of the proposed system, this approach will open up a number of bio-analytical opportunities in applications from molecular recognition, classification/identification of biological samples to disease diagnosis and cellular monitoring.

## Methods

**Chemicals and materials**. Polystyrene (300 nm diameter), Triton X-100, 4-aminophenylacetic acid, 1-undecanethiol, 1-propanethiol, 11-mercaptoundecanoic acid, 3-mercaptopropionic acid, 11-mercapto-1-undecanol, 3-mercapto-1-propanol, 11-amino-1-undecanethiol hydrochloride, 3-amino-1-propanethiol hydrochloride, Rhodamine 6G, 4-aminophenlyacetic acid, folic acid, $p$-phenylenediamine, acetic acid, and ammonium hydroxide were supplied by Sigma-Aldrich (St. Louis, MO, USA). Pure anhydrous ethanol (VWR, Radnor, PA, USA) and nuclease-free water (Invitrogen, Carlsbad, CA, USA) were used in all preparation. All the chemicals were used without further purification.

**Au-nanopillar fabrication**. The fabrication consists of four stages: (i) deposition of low-stress silicon nitride ($Si_3N_4$, 120 nm) on Si wafer, (ii) spin coating of polystyrene (PS) beads on the wafer, (iii) reactive ion etching (RIE) to tailor the PS beads and etch the pillar structures, and (iv) thermal deposition of Cr/Au layer. For the $Si_3N_4$ deposition, low-pressure chemical vapor deposition (LPCVD) was used over 100 mm, 0.01 Ω-cm, boron-doped p-type silicon wafers. Before spin coating, the coated wafer was cut into $2 \times 2$ cm$^2$ pieces and cleaned by $O_2$ plasma cleaning for 5 min (Electronic Diener, LFG 40). Then PS bead solution (3.0−3.75 wt% in 0.25 wt% Triton X-100-containing absolute ethanol) was spin-coated with a spin coater (CEE Apoge) to form a monolayer. The spin-coating programme consisted of three stages: (i) 500 rpm for 10 s with acceleration of 100 rpm/s, (ii) 1200 rpm for 15 s with acceleration of 400 rpm/s, and (iii) 2500 rpm for 10 s with acceleration of 1000 rpm/s. Further tailoring of beads and the nitride etching step were done by parallel plate RIE etcher (Plasma Pro NGP80) with 50 sccm of $CHF_3$ and 5 sccm $O_2$ at a pressure of 55 mTorr and radio frequency (RF) power of 150 W. Etch time was 2.5 min to achieve full etch of the nitride layer. After the RIE, Cr (10 nm) and Au (70 nm) was thermally evaporated consecutively (Edwards A306 Box Evaporator) to cover the nanopillar structure. Deposition rates for each metal were 0.1 and 0.5 Å/s, respectively. The base pressure for the metal layer deposition was lower than $10^{-5}$ Torr. All steps were done in a Class 2 cleanroom to avoid contamination.

**Self-assembled monolayer (SAM) modification of Au-nanopillars substrates**. Prior to functionalization, the fabricated Au-nanopillars were cleaned via a

reported two-step protocol[64] with adaptations: oxygen plasma treatment to remove organic contaminants and immersion in pure ethanol to reduce resultant gold oxide layers. Alkyl- and hydroxyl-terminated SAMs were formed by immersion of the cleaned Au-nanopillars substrates in 1 mM solution in pure ethanol at room temperature for 24 h. For amine-terminated SAMs, 10% ammonium hydroxide (v/v) was added in the 1 mM ethanol solution followed by immersion of the substrates for 24 h. Previous reports have reported that a basic ethanolic solution improves packing of amine-terminated SAMs by reducing the presence of unbound thiol molecules that are susceptible to oxidation[65]. For carboxyl-terminated SAMs, 10% acetic acid (v/v) was added in 1 mM of ethanol solution, and the substrates were immersed for 24 h. Upon removal, the substrates were rinsed sequentially with pure ethanol, distilled water, and pure ethanol followed by blow drying in a nitrogen stream[65,66].

**Preparation of extracellular vesicles**. Cell-derived (extracellular) vesicles were prepared from MDA-MB-231 breast cancer cells, obtained from the American Type Culture Collection (ATCC), HTB-26. Cells were cultured for 72 h in serum-free Dulbecco's modified Eagle's medium 37 ℃, 5% $CO_2$. The conditioned medium was collected and concentrated using ultrafiltration against a 100 kDa regenerated cellulose membrane. The concentrate was subjected to size exclusion chromatography using a resin of Sepharose CL-2B (Sigma). Column fractions containing EVs were collected and stored at −80 ℃ until further analysis.

**Preparation of cell lysates**. Hs578Bst and Hs578T cells were purchased from the ATCC, HTB-125 and HTB-126, respectively. Cells were maintained in the growth media advised by ATCC. For preparation of cell lysates, cells were grown to around 90% confluence in T75 cell culture flasks. Cell culture medium was aspirated, cells were washed once with PBS to ensure a controlled pH (10 mM, pH 7.4) (Life Technologies) and harvested using scraping. The cell suspension was transferred to microcentrifuge tubes and sonicated to release the intracellular content with a probe sonicator (Cole-Parmer) at 20% amplitude in 03 cycles of 15 s each, with a 5 s break in between. Cell lysates were aliquoted and stored at −80 ℃ until further analysis.

**SERS measurements**. SERS spectra were recorded using a Renishaw Invia Raman microspectrometer. The instrument consists of a Leica microscope, a grating of 1200 groove/mm and a Peltier cooled NIR optimized CCD detector. Raman measurements were performed by coupling a 785 nm NIR diode laser via a ×50/ 0.75 NA Leica objective lens. Spectra were collected with laser powers between 5 and 13 mW with the CCD exposure times of 5–0 s depending on the samples. All data processing was performed using MATLAB (MathWorks, Inc., Natick, MA) and the PLS-Toolbox (Eigenvector Research, Inc., Manson, WA). The spectral smoothing was performed using the Savitsky–Golay method with a second-order polynomial and window size of 9[33]. The baselines were subtracted with a quadratic fit using the 'msbackadj' algorithm, which is the baseline removal function from MATLAB[67]. For four molecular target signatures, due to varying width of peaks, a variable window from 50 to 165 and step size from 15 to 30 were implemented to remove the background with minimal effect on the SERS peaks. A set window size (120) and step size (35) were used for all cell lysates SERS data processing to eliminate the potential risk of biasing the statistical multivariate analysis between samples. SERS mapping was performed on a confocal Raman micro-spectroscope (alpha300R+, WITec, Ulm, Germany) using a 785 nm laser (Toptica XTRA II) of 10 mW laser power with the application of a ×50/0.75 NA microscope objective lens (Carl Zeiss Microscopy, Oberkochen, Germany). The scattered light was guided via a 100 μm fibre with a 600 groove/mm grating spectrograph (UHTS 300, WITec, Ulm, Germany) and spectra were acquired using a thermoelectrically cooled back-illuminated CCD camera (iDus DU401-DD, Andor, Belfast, UK). The spectra were pre-processed using WITec Control FOUR software (v. 4.1) being cropped to the range of 727–1782 $cm^{-1}$, smoothed using the Savitsky–Golay method with a second-order polynomial, and baseline corrected via curve fitting using the built-in shape filter (shape size = 200). Subsequent spectral processing was performed using MATLAB (MathWorks, Inc., Natick, MA) where the spectra were normalized by area under the curve according to the approach employed for biological sample measurements and prominent peak area was used for mapping.

**Characterization of the FASERS**. The morphology of the fabricated FASERS was evaluated by high-resolution field emission gun scanning electron microscopy (FEGSEM) using Zeiss Sigma 300 FEGSEM. SERS activity of the unfunctionalized Au-nanopillars substrates was investigated with chemisorbed 4-aminothiophenol (ATP). The cleaned Au-nanopillars were immersed in an ATP ethanol solution of varied concentration from 100 nM to 10 mM for 24 h. The SERS spectra were recorded with laser power of 5 mW with CCD exposure of 5 s. The measurement was carried out on three randomly selected points using three different substrates ($N = 3$, $n = 3$). Each spectrum was normalized by the standard silicon peak (~520 $cm^{-1}$), smoothed, baseline subtracted, and averaged. To perform SERS mapping of the substrate, cleaned Au-nanopillars and Au–Si substrates were immersed in a 1 μM 4-mercaptobenzoic acid (4-MBA) ethanolic solution for 24 h. The prominent peak of 4-MBA centred at 1074.7 $cm^{-1}$ (sum of the peak intensity between 1057 and 1092 $cm^{-1}$) was used for SERS mapping (50 μm × 50 μm, pixel

size of 1 $μm^2$, integration time 2 s). Upon SAM functionalization, a contact angle goniometer was used to evaluate the formation of SAM using 3 μL droplet of nuclease-free water (Invitrogen). AFM imaging was carried out on the functionalized SERS substrates using a Keysight Technologies 5500 AFM with a Mikro-Masch HQ:NSC cantilever (nominal spring constant of 40 N/m). Three regions of interest (4 $μm^2$) were imaged for each sample in an ambient atmosphere. The mean roughness ($R_a$) values were obtained using Gwyddion software (http://gywddion. net). The XPS measurements were performed on functionalized SERS substrates using a Thermo Fisher K-Alpha+ spectrophotometer. This equipment employs a monochromatic Al-Kα X-ray source and a 180° double focusing hemispherical analyser with a 2D detector. Using a 400 $μm^2$ X-ray spot size, survey spectra were recorded for each sample at pass energies of 200 eV followed by high-resolution measurements for Au 4f, S 2p and C 1s core levels at pass energies of 20 eV. For each sample, a flood gun was employed to minimize sample charging during photoelectron collection. Curve fitting and data analysis was carried out using the Avantage Software package (v.5.9490). The principal C 1s peak, the band at 284.8 eV, was used as the internal standard for charge correction, and each spectrum was normalized to the maximum intensity. The SERS spectra of FASERS in PBS (pH 7.4) environment were obtained with laser power of 5 mW and acquisition time of 10 s. For each sample, the measurement was carried out on three randomly selected points using three different substrates ($N = 3$, $n = 3$). Each spectrum was smoothed, baseline subtracted and normalized by the area under the curve.

**Detection of small molecules and biological samples on FASERS**. The small molecule detection was carried out by loading 5 μL droplets of molecular solutions onto each differently functionalized Au-nanopillar surface. p-PDA ($pK_a = 6.04$) and R6G ($pK_a = 7.50$) were dissolved in phosphate buffer (10 mM, pH 5.0), and 4-APA ($pK_a = 3.83$) and FA ($pK_a = 3.37$) were dissolved in phosphate buffer (10 mM, pH 7.5). Given these pH and $pK_a$ values, p-PDA and R6G should have been in a predominantly protonated state, and 4-APA and FA in a deprotonated state in each buffer condition. With respect to the SAMs, the surface $pK_a$ (the dissociation constant, $pK_d$) of the carboxyl and amine-terminations have been reported for 3-mercaptopropionic acid (surface $pK_a$ ~ 5.2[68]), 11-mercaptondecanoic acid (surface $pK_a$ ~ 5.0[69]), 3-amino-1-propanethiol (surface $pK_a$ ~ 8.5[69]) and 11-amino-1-undecanethiol (surface $pK_a$ ~ 8.9[70]). As a result, the carboxyl end-groups are expected to be largely deprotonated at pH 7.5 and in the mixed state of deprotonated/neutral at pH 5.0. On the other hand, the amine end-groups can be assumed to be largely in the protonated state in both pH 7.5 and pH 5.0. A laser power of 5 mW for the acquisition times of 5 s (p-PDA) and 10 s (4-APA, FA, R6G) were used during measurements. For each sample, the measurement was carried out on three randomly selected points using three different substrates ($N = 3$, $n = 3$). Each spectrum was normalized by the standard silicon peak (~520 $cm^{-1}$), smoothed, baseline subtracted, and averaged. For biological samples, an aliquot of cell lysate was freshly defrosted prior to measurement and 5 μL droplet placed on each self-assembled monolayer functionalized substrate. The SERS spectra were recorded at each different substrate on two randomly selected points using three different substrates batches ($N = 3$, $n = 2$) to ensure the reproducibility of the data. The laser power of 13 mW for an acquisition time of 10 s was used for the measurements. All the spectra were smoothed, baseline subtracted, and subsequently normalized to the area under the curve, which removed any instrumental effects and enabled comparisons between the samples by reducing the variability of signal intensity.

**Multivariate analysis**. Multivariate analysis was performed using MATLAB (MathWorks, Inc., Natick, MA, USA) and the PLS-Toolbox (Eigenvector Research, Inc., Manson, WA, USA). Principal component analysis was performed individually on the dataset obtained from each functionalization after mean centreing. The first principal component (PC1) was obtained for each functionalization, resulting in a total of nine PC1, and used for high-dimensionality analyses. The nine PC1s were cross-combined for different number of axes at each dimensionality (1–9), followed by linear discriminant analysis for discrimination of the two cell lysates. Considering all the possible combinations, in total 511 PCA–LDA models were calculated: 9 ($x = 1$), 36 ($x = 2$), 84 ($x = 3$), 126 ($x = 4$), 126 ($x = 5$), 84 ($x = 6$), 36 ($x = 7$), 9 ($x = 8$), 1 ($x = 9$) where $x$ refers to the number of SAMs used for analyses at each dimensionality. The prediction performance was evaluated for all the PCA–LDA models using leave-one-out cross-validation method (LOOCV). The predictive performance of the cross-validated models was assessed by the misclassified fraction of the observations out of all the validated observations, defined as the term 'cross-validated classification error'. Linear discriminant analysis was carried out using MATLAB in-built LDA model training function and label-prediction function. The cross-validated models were obtained using MATLAB in-built function for leave-one-out cross-validated discriminant analysis. The cross-validated classification error was calculated using the MATLAB in-built function while the loss function was specified as a classification error. The confidence interval boundaries were obtained with OriginLab software OriginPro2019 (OriginLab Corporation, Northampton, MA, USA).

## Data availability

Experimental raw data are available at https://doi.org/10.5281/zenodo.3525182 and molecular dynamics simulation data from irene.yarovsky@rmit.edu.au.

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

## Acknowledgements

N.K. acknowledges support from the Korean Government Scholarship for Study Overseas funded by the Korean Ministry of Education. M.R.T. and M.M.S. acknowledge support from the i-sense Engineering and Physical Sciences Research Council (EPSRC) IRC in Early Warning Sensing Systems for Infectious Diseases (EP/K031953/1; EP/R00529X/1; www.i-sense.org.uk). M.R.T. and M.M.S. acknowledge support from the Medical Research Council (MRC) grant "m-Africa" (MR/P024378/1). M.S.B. acknowledges support from the European Union's H2020 Research and Innovation programme under the Marie Skłodowska-Curie Fellowship "IMAGINE" (701713). I.J.P. acknowledges support from the Whitaker International Program, Institute of International Education, United States of America. H.S. acknowledges support from the Basic Science Research Program through the National Research Foundation of Korea (NRF) funded by the Ministry of Education (2017R1A6A3A03007397) and the European Union's H2020 Research and Innovation programme under the Marie Skłodowska-Curie Fellowship "ELECTRO NEEDLE" (797311). A.N. acknowledges support from the National Institute for Health Research (NIHR) Imperial Biomedical Research Centre and the Institute of Cancer Research, London, through the joint Cancer Research Centre of Excellence (CRCE). A.G. acknowledges support from the European Union's Horizon 2020 Research and Innovation Programme through the Marie Skłodowska-Curie Individual Fellowship "RAISED" (660757). D.J.P. acknowledges support from the Royal Society (Grants No. UF100105 and No. UF150693) and EPSRC (Grant No. EP/M028291/1). I.Y. and M.M.S. acknowledge the Australian Research Council Discover grants DP140101888 and DP170100511. A.B.R. acknowledges a studentship from the EPSRC CDT for the Advanced Characterisation of Materials (EP/L015277/1). P.C., N.T. and I.Y. acknowledge grant e87 for access to high performance computational resources provided by the Australian government via the NCI and Pawsey facilities. All authors would like to thank Seung Kyun Ha for helpful discussions regarding statistical analysis and its practical implementation.

## Author contributions

N.K., M.R.T., and M.S.B. conceived the project. N.K. and M.R.T. designed most of the experiments. N.K., M.R.T. and M.M.S. wrote the paper. N.K. performed the majority of experiments and data processing. H.S. manufactured plasmonic gold nanopillar substrates. P.C., N.T. and I.Y. performed molecular dynamics simulations. N.K., M.S.B. and I.P. conceived and implemented the statistical analysis approaches. A.B.R. and D.P. performed and interpreted XPS measurements. A.N. prepared cell lysates. A.G. performed AFM analyses. All the authors discussed the results and assisted in the preparation of the paper. M.M.S. also contributed to study design and supervised the project.

## Competing interests

The authors declare no competing interests.
