## [Peer Review File · Nature Communications]

Reviewers' comments:

Reviewer #1 (Remarks to the Author):

In this work, authors reported the development of an artificial-nose inspired SERS fingerprinting approach for the discrimination of complex biological samples, specifically, Hs578T breast carcinoma cells and Hs578Bst normal fibroblast-like cells, based on arrays of 8 different self-assembled monolayers (SAMs) functionalized SERS-active substrates. 8 alkane thiol molecules were selected with four of them having 3 carbon chain length and another four having 11 carbon chain length, each set with four functional groups: -CH₃, -OH, -NH₂, and -COOH, to represent possible hydrophobic interaction (-CH₃ and -OH) and electrostatic interaction (-NH₂ and -COOH) with analytes. SAMs with different chain lengths can not only physically separate analytes from the plasmonic gold surface but also could provide different packing density and conformation for analyte partitioning in SAMs to enhance selectivity and SERS signal. Authors took the SERS spectra of each SAM and characterize the quality of SAM using XPS. Authors first demonstrated that 4 simple molecules: p-phenylenediamine (p-PDA), 4-aminophenylacetic acid (4-APA), rhodamine 6G (R6G), and folic acid (FA), present different SERS spectra on these 8 SAMs surfaces, indicating selective interactions. Further detailed studies were carried with p-PDA and 4-APA under different pH conditions both experimentally and with molecular simulations. Simulations predict/confirm the orientation and penetration of these two molecules on 8 SAMs under different pH conditions, which support the experimental observation. Finally, authors collected SERS spectra of normal and cancer cell lysates on these 8 SAM functionalized SERS substrates and applied comprehensive multivariate analysis to demonstrate the selectivity, specificity and accuracy on the discrimination of normal and cancer cell lysates with the combination of bare and surfaces functionalized with different SAMs (both chain length and function groups). The work is complete and the selective responses of different SAM functionalized SERS surfaces to analytes are interesting. The work would be of immediate interest to researchers in SERS biosensing and diagnostics. It is recommended for publication on Nature Communications.

Reviewer #2 (Remarks to the Author):

In this manuscript, Stevens et al. report a SERS based detection system. Using an interfacial array of 8 chemically different surfaces, the manuscript pursues an unbiased detection approach. Similar ideas have been previously developed for other analytical methods, such as Mass Spec, but the work shown here on the basis of Raman spectroscopy is original and impactful. This is a very strong paper from one of the leading research groups in the field. The DFT calculations add further strength and complement the unbiased approach nicely.

In general, this is a very well written manuscript with high-quality data and the conclusions are supported by a combination of experimental, analytical and theoretical work. The paper should be accepted after a few minor comments are addressed:

1) For any viable analytical technique, specificity and selectivity are critical. The authors demonstrate that they can distinguish between two different cell lysates, cancer vs. non-cancer. However, can they also identify samples from different cell lysates of distinct cancer cell populations as similar? With other words, if the point is to detect that someone has cancer, it will be as important to identify common similarities between let's say cancer patients, as it is to distinguish between two cell populations.

2) The computer simulations in Fig. 4 are important as the attempt to validate some of the unbiased studies, but (at least for me) are impossible to discern. The details are too small and can't be read. It is also not entirely clear what the angles mean here. At least one of the analytes, PDA, is symmetric and so how is the angle defined here? The authors may want to consider comparing it to aniline or something similar as well.

3) In Figure 2c, the signals for the different monolayer are very different. In particular $\text{CH}_3(\text{CH}_2)_2\text{-SH}$ appears very strong. The authors should confirm that this spectrum indeed is consistent with the chemical structure of the thiol.

Reviewer #3 (Remarks to the Author):

In this work, authors report a Functionalized Array SERS platform (FASERS) for the detection of small molecule analyte/bio-samples.

Authors presented the work in a detailed way with lots supporting information data. Though this work is interesting, but it lacks the novelty and innovativeness to warrant publication in Nature comm.

My concerns are given below

1. Label free SERS is only applicable when the target molecule has inherently large Raman cross section. However this is not true for many biologically relevant cases for small molecules (such as glucose) or biomarker detection (proteins) . So by doing FASERS (basically nothing advanced but a modified Self assembled monolayer/multilayer (SAM), the fundamental issue with label free SERS could not overcome to a great extent.

2. Most significantly, authors adopted FASERS to differentiate the breast carcinoma cells with normal fibroblast-like cells from the same individual by employing established statistical tool like PCA/PCA-LDA. There is no advancement in this aspect of the work. First of all, authors should have studied large sets of samples that has statistical significance (at least 20-30 cancer and may be same number of healthy samples). Comparing the data from 1 set of samples is not acceptable and wont convince any. There are many reports on simple label free SERS with statistical classification that resulted in high accuracy (sensitivity/specificity) in the range of over 90%. (some of these works are listed below). In this context, the advancement made using FASERS is quite minimal.

i) Biosensors and Bioelectronics 25, 2414–2419 (2010)

ii) Analyst, 138, 3967 (2013)

iii) Optics Express 19 (14), 13565-13577, (2011)

Differentiating normal and cancer cells (lysate) is far easier than differentiating different types of closely associated cancers. If FASERS provides high specificity, authors should have demonstrated it to establish its superiority. As indicated in above references, differentiating cancer Vs Normal directly from patient's serum (or body fluids) is significant and has much more clinical value than testing it in the cell lysate as demonstrated here.

With FASERS approach, I assume it is really difficult to achieve highly sensitive detection in a real complex body fluids (such as blood or saliva) by overcoming the biofouling. Authors should have demonstrated it in this paper.

3) For any practical applications, specifically for bio medical, if the analyte of interest has weak Raman cross section, targeted detection (using targeting moieties like antibodies/aptamers or using SERS nanotag with a reporter molecule) is always well accepted and had shown highly promising results. In this context, the label free approach with "self-assembled layer" is not innovative or revolutionary.

4) As reported, if this work has the advancement as "artificial-nose based sensing methodology", authors should have demonstrated it in a various complex media that include biological/non-biological.

Also, establishing the limit of detection that breaks the current limit for some of the trace biochemicals would have attracted great interest

Even establishing the detection of glucose in a physiologically relevant condition should have attracted interest.

5) SERS substrate reported here is also not novel. There are many reports on similar SERS substrates.

It would have been nice if the SERS mapping is demonstrated to establish the reproducibility in signal at various spatial location of the substrate.

Response to All Reviewers

We thank the reviewers for their constructive and insightful comments, which we have addressed in detail in this point-by-point response. *The reviewers' comments are in black with authors point-by-point responses in blue, and changes to the manuscript highlighted in yellow.* Changes to the manuscript are highlighted in yellow in the revised main text and Supplementary Information files, and the methods section in the manuscript has been updated to reflect the new figures.

Reviewer #1 (Remarks to the Author):

In this work, authors reported the development of an artificial-nose inspired SERS fingerprinting approach for the discrimination of complex biological samples, specifically, Hs578T breast carcinoma cells and Hs578Bst normal fibroblast-like cells, based on arrays of 8 different self-assembled monolayers (SAMs) functionalized SERS-active substrates.

8 alkane thiol molecules were selected with four of them having 3 carbon chain length and another four having 11 carbon chain length, each set with four functional groups: -CH₃, -OH, -NH₂, and -COOH, to represent possible hydrophobic interaction (-CH₃ and -OH) and electrostatic interaction (-NH₂ and -COOH) with analytes. SAMs with different chain lengths can not only physically separate analytes from the plasmonic gold surface but also could provide different packing density and conformation for analyte partitioning in SAMs to enhance selectivity and SERS signal. Authors took the SERS spectra of each SAM and characterize the quality of SAM using XPS.

Authors first demonstrated that 4 simple molecules: p-phenylenediamine (p-PDA), 4-aminophenylacetic acid (4-APA), rhodamine 6G (R6G), and folic acid (FA), present different SERS spectra on these 8 SAMs surfaces, indicating selective interactions. Further detailed studies were carried with p-PDA and 4-APA under different pH conditions both experimentally and with molecular simulations. Simulations predict/confirm the orientation and penetration of these two molecules on 8 SAMs under different pH conditions, which support the experimental observation.

Finally, authors collected SERS spectra of normal and cancer cell lysates on these 8 SAM functionalized SERS substrates and applied comprehensive multivariate analysis to demonstrate the selectivity, specificity and accuracy on the discrimination of normal and cancer cell lysates with the combination of bare and surfaces functionalized with different SAMs (both chain length and function groups).

The work is complete and the selective responses of different SAM functionalized SERS surfaces to analytes are interesting. The work would be of immediate interest to researchers in SERS biosensing and diagnostics. It is recommended for publication on Nature Communications.

We greatly appreciate the reviewers support of our work.

Reviewer #2 (Remarks to the Author):

In this manuscript, Stevens et al. report a SERS based detection system. Using an interfacial array of 8 chemically different surfaces, the manuscript pursues an unbiased detection approach. Similar ideas have been previously developed for other analytical methods, such as Mass Spec, but the work shown here on the basis of Raman spectroscopy is original and impactful. This is a very strong paper from one of the leading research groups in the field. The DFT calculations add further strength and complement the unbiased approach nicely.

In general, this is a very well written manuscript with high-quality data and the conclusions are supported by a combination of experimental, analytical and theoretical work. The paper should be accepted after a few minor comments are addressed:

We thank the reviewer for their support and for taking the time to provide detailed constructive feedback on our manuscript and address their comments in full below.

1) *For any viable analytical technique, specificity and selectivity are critical. The authors demonstrate that they can distinguish between two different cell lysates, cancer vs. non-cancer. However, can they also identify samples from different cell lysates of distinct cancer cell populations as similar? With other words, if the point is to detect that someone has cancer, it will be as important to identify common similarities between let's say cancer patients, as it is to distinguish between two cell populations.*

Above all, we thank the reviewer for raising additional scope in the potential applications of our platform. We have understood the reviewer's question as being whether our FASERS system identify common similarities among different cancer cell lysates (e.g. breast cancer, lung cancer, prostate cancer, etc.) between patient populations.

Indeed, there are a variety of distinct biochemical signatures that malignant cells commonly exhibit, such as increased synthesis of RNA and DNA, elevated level of lactic acidosis, aberrant glycosylation, and alteration in compositions of cell surface glycoproteins, proteoglycans, glycolipids and mucins¹. Although the scope of our current work is to demonstrate the merit of the approach of hypothesis-free discrimination, we believe both the discrimination and the detection of common features can be assisted by the same principle of amplifying the information dimensionality using an array of midly-selective plasmonic surfaces. To be more specific, supposing some of the functionalities effectively interrogate common compositional aspects of malignant cells providing specific SERS spectral features, the summation of corresponding principal components (PCs) will provide more effective separation of normal cells from all-other-cancer cells in PC space by co-clustering different types of cancer cells as similar, supported by the common features. We propose this expansion of the applications could be realized by careful selection of SAMs and the potential implementation of machine-learning-based feature selection algorithms, which could represent an important follow-up research direction.

Highlighting the reviewer's suggestion, we have included the following in the revised text:

Revision made:

- **Text in the manuscript page 22** has been revised accordingly:

This implementation of an artificial-nose empowered SERS approach to generate and analyze high dimensionality data by increasing the number of spectral outputs from a single sample represents a proof-of-concept artificial-nose based sensing methodology. The system represents a hypothesis-free approach that could be exploited towards accurately discriminating the differences between distinct

populations but also to identify the common similarities among closely-related populations. This offers a new platform technology to characterize, cluster and classify biological samples, which in the context of diseases, such as cancer, opens up numerous possibilities for a simple yet powerful label-free diagnostic platform.

2) *The computer simulations in Fig. 4 are important as the attempt to validate some of the unbiased studies, but (at least for me) are impossible to discern. The details are too small and can't be read. It is also not entirely clear what the angles mean here. At least one of the analytes, PDA, is symmetric and so how is the angle defined here? The authors may want to consider comparing it to aniline or something similar as well.*

We thank the reviewer for their comment and value the feedback that they find the details of Figure 4 difficult to interpret, thus we have revised and clarified the manuscript accordingly. The role of this two-part figure is as follows: Figure 4a depicts representative snapshot images to visually highlight the smallest separation distance visited in the simulations between the gold surface/SAM and the p-PDA/4-APA analytes, as well as to emphasise the orientation adopted by the molecules; whereas Figure 4b displays the data for the relative ensemble of orientations each molecule assumes when in proximity to the SAM/surface.

To address the reviewer's comment about the definition of the orientation angles represented in Figure 4b, we refer to the figure caption which explains the purpose and meaning of the orientation angles and reads as: "*Analyte orientation angles of (b) p-PDA and (c) 4-APA relative to the Au surface showing the direction of each analyte's functional groups when proximal (< 0.6 nm) to a SAM/Au. Angles < 90° indicate that the NH₂ group of the analyte is pointing towards the SAM/Au surface, whereas angles > 90° specify that the charged groups (NH₃⁺/COO⁻) are facing the SAM/Au.*" Further to this, the supporting information on page 14 provides the exact definition of the orientation angles and reads as: "*The analyte molecular orientation relative to Au(111) surface, ϕ , was measured as the angle between the analyte molecular vector and the gold substrate normal vector. For p-PDA, the molecular vector was defined between nitrogen on NH₂ and nitrogen on NH₃⁺, while for APA this was between nitrogen on the NH₂ and carbon on COO⁻.*" These vectors and angles are also defined in explicit detail in Figure S5.

As correctly identified by the reviewer, p-phenylenediamine is a symmetric molecule when both amine groups possess a neutral charge and therefore a unique molecular vector (and angle) would be impossible to assign; however, as described above, and noted in our Computational Details on page 14 of Supplementary Information, we have simulated the asymmetric cationic form of p-PDA. To further emphasise this and improve the ability for readers to discern the overall molecular details presented in Figure 4, we have: updated the chemical structures displayed in the far right-hand column of Figure 4a to reflect the protonation states of the p-PDA and 4-APA analytes and recolored oxygen atoms (red) and nitrogen atoms (blue) to correspond with the rendered molecular snapshots; likewise, we have included the location of the analytes' charged functional groups in the snapshots of Figure 4a; and added the following to the final line of the Figure 4 caption:

"SAM and analyte protonation states are modelled based on the cognate experimental buffer conditions, i.e. p-PDA (pH 5) and 4-APA (pH 7.5)."

We believe these changes improve the clarity of the figure and help to appropriately direct readers to interpret the information presented.

We appreciate the suggestion to compare with aniline. Indeed, there are a number of closely matched compounds that could be interesting to investigate and understand the impact of subtle variations in molecular structure. As described on page 11 of the manuscript, "*In order to demonstrate SAM-dependent SERS output signatures from each receptor, we [experimentally] measured four model molecular solutions of differing charge, size and*

hydrophobicity by FASERS ... To obtain insight into the underlying interactions at play, we [computationally] investigated the two closely matched small molecules with opposite electrostatic potential, 4-APA and *p*-PDA at each SAM interface with all-atom molecular dynamics (MD) simulations". Aniline (pKa 4.6), in both the neutral and cationic forms, does not bear any functional group diversities from the small molecules investigated and thus we do not believe such simulations will provide significant additional information for understanding the substrate interactions of the *p*-PDA and 4-APA analytes already simulated and supported by the experimental data.

Revision made:

- Figure 4 in the manuscript page 15 has been revised accordingly:

- Figure 4 caption in the manuscript page 15 has been revised accordingly:

Figure 1. Molecular dynamics simulation results for *p*-phenylenediamine (*p*-PDA) and 4-aminophenylacetic acid (4-APA) molecules on SAM-functionalized gold surfaces. (a) Snapshots of molecules where the analyte-gold surface separation reaches a minimum. Analyte center-of-mass is used to measure the distance to the closest Au surface atom; % populations when analytes proximal (< 0.6 nm) to a SAM/Au are calculated from MD generated equilibrium ensemble for each system: non-functionalized (bare), 1-propanethiol (3CH₃), 3-mercaptopropanol (3OH), 3-mercaptopropionic acid (3COOH), 3-amino-1-propanethiol (3NH₂), 1-undecanethiol (11CH₃), 11-mercaptopropanol (11OH), 11-mercaptopropionic acid (11COOH), 11-amino-1-undecanethiol (11NH₂). Analyte orientation angles of (b) *p*-PDA and (c) 4-APA relative to the Au surface showing the direction of each analyte's functional groups when proximal (< 0.6 nm) to a SAM/Au. Angles < 90° indicate that the NH₂ group of the analyte is pointing towards the SAM/Au surface, whereas angles > 90° specify that the charged

groups ($\text{NH}_3^+/\text{COO}^-$) are facing the SAM/Au. SAM and analyte protonation states are modelled based on the cognate experimental buffer conditions, i.e. *p*-PDA (pH 5) and 4-APA (pH 7.5).

3) In Figure 2c, the signals for the different monolayer are very different. In particular $\text{CH}_3(\text{CH}_2)_2\text{-SH}$ appears very strong. The authors should confirm that this spectrum indeed is consistent with the chemical structure of the thiol.

We thank the reviewer for their important question about the spectral fingerprints of monolayers. To address this query, we have further described the tentative assignment of the prominent peaks for each monolayer, and the results are now included in the new supplementary figures and additional text added to the manuscript as outlined below.

In summary, the C-C stretching region ($1000\text{--}1500\text{ cm}^{-1}$) shown in the figure contains important information about the conformational behaviour of the alkyl chain, arising from the *gauche* and *trans* conformers of C-C bonds. The positions of prominent peaks in SERS signatures agree well with a number of previous studies^{2, 3, 4, 5, 6, 7, 8, 9, 10, 11}, being consistent with the chemical structures of the SAM-forming molecules used. Regarding the relatively strong intensity of (C-C) stretching of 1-alkanethiol compared to other thiolates, this is attributed to the molecules having the lowest symmetry along the vibration axis and the vibrations of the adsorbed molecule have a component perpendicular to the surface in all orientations^{3, 5}. As the vibration components along the z-axis (i.e. zz tensor component) are enhanced to a larger extent than those with xz or yz components, the C-C components of 1-alkanethiol show a large contribution of the stretching vibration of the *trans*-conformer, which agrees well with reported literature^{3, 5}.

Revision made:

- **Text in the manuscript page 7–8** has been revised accordingly:

We fabricated SERS-active gold film coated nanopillar substrates (Au-nanopillars) via colloidal lithography and plasma etching as described in Figure 2a and Figure S1a–b (Supplementary Information). We demonstrated SERS activity of the fabricated Au-nanopillars with 4-aminothiophenol (ATP) that was chemisorbed onto the surface via Au-S bonds. We did not detect any significant SERS signatures for the bare Au-nanopillars while SERS signatures with varying degrees of intensity were detected for the functionalized Au-nanopillars. In particular, relatively strong signatures were obtained from the 1-alkanethiols, consistent with reported literature^{3, 4, 5, 6}, which is attributed to the lowest symmetry along the vibration axis and large contribution from stretching of *trans*-conformers. The four ligands with 11-carbon alkyl chains showed comparable SERS signatures with a predominant peak at about $1100\text{--}1200\text{ cm}^{-1}$, which corresponds to symmetric the C-C stretching mode being the largest contribution from the tensor component along the axis of the vibration^{3, 5}. Detailed tentative assignments of the peaks are further discussed in Supplementary Information Table S1.

- **Text and Table S1 in the Supplementary Information section II.3** has been added:

II.3. Tentative peak assignment of SERS spectra from self-assembled monolayers

The C-C stretching region ($1000\text{--}1500\text{ cm}^{-1}$) of the monolayers contains information about the conformational behaviour of the alkyl chain, arising from the *gauche* and *trans* conformers of C-C bonds.

Above all, the prominent peaks around $1100\text{--}1130\text{ cm}^{-1}$ are corresponding to (C-C) *trans*-conformers, the positions of which agree well with a number of previous studies².

^{3, 4, 5, 6}. Regarding the relatively strong intensity of (C-C) stretching of 1-propanethiol and 1-undecanethiol compared to other short thiolates, this is attributed to the 1-alkanethiol molecules having the lowest symmetry amongst the C_s point group, therefore the vibrations of the adsorbed molecule exhibit a component perpendicular to the surface in all orientations^{3, 5}. As the vibration components along the z-axis (i.e. zz tensor component) are enhanced to a larger extent than those with xz or yz components, the C-C components of 1-alkanethiol show a large contribution of stretching vibration of *trans*-conformers^{3, 5}. On the other hand, the short-chain alkanethiol with a substituted terminal group can also interact with the metal surface^{4, 7, 8, 9, 10, 11}. In particular, the terminal carboxylic and amino group are known to exhibit a relatively higher concentration of *gauche* conformers via double bonding to the metal surface^{4, 7, 8, 9}, which leads to minimized stretching vibration of *trans*-conformers when compared to 1-propanethiol.

For monolayers with 11 carbon atoms (11CH₃, 11OH, 11COOH, 11NH₂), the peak corresponding to stretching vibration of the C-C *trans*-conformer (i.e. $v_s(C-C)_T$) are the most clearly observable of the signals. This is consistent with the findings of previous studies in which monolayer structures begin to resemble that of the solid from carbon number 8⁶, and the symmetric stretching vibrations of C-C bonds are the largest contribution from the tensor component along the axis of the vibration⁵.

Table S1. Tentative assignments of prominent peaks of SAM monolayers.

SAM-forming molecules	3CH ₃	3OH	3COOH	3NH ₂	11CH ₃	11OH	11COH	11NH ₂
Raman shift (cm⁻¹)								
^{a,b} CH ₂ (rock) _G ^{5, 6}	838				860.0			
^{a,b} CH ₃ (rock) _T ^{5, 6}	889 1002				891 1004	999	1003	1004
$v(C-COO^-)$ ^{10, 11}			921					
^a $v(C-C)_T$ ^{5, 6}	1028	1028						
^a $v(C-C)_G$ ⁵			1042		1065 1074	1062	1062	1096
^{a,c} $v_s(C-C)_T$ ^{3, 6}					1125	1108	1099	1107
^{a,c} $v_s(C-C)_T$ ^{3, 5, 6}	1087			1119	1125	1108	1099	1107
^b CH ₂ wag ^{5, 6}	1283.8	1248		1281	1297	1298	1272 1301	1258
^{b,d} CH ₃ sy.def ⁶	1324 1384 1415	1384	1384 1415	1384 1437	1323 1344 1384 1437	1324 1345 1358 1383 1397	1322 1345 1362 1384 1396	1304 1321 1344 1383
^{b,d} CH ₃ as.def ⁶	1454		1446	1448	1449	1437	1447	1442

^a ()_G: *gauche*-conformer, ()_T *trans*-conformer. ^b def.: deformation, wag.: wagging, rock: rocking. ^c v_s : stretching. ^d sy.: symmetric, as.: asymmetric.

Reviewer #3 (Remarks to the Author):

In this work, authors report a Functionalized Array SERS platform (FASERS) for the detection of small molecule analyte/bio-samples.

Authors presented the work in a detailed way with lots supporting information data. Though this work is interesting, but it lacks the novelty and innovativeness to warrant publication in Nature comm.

We thank the reviewer for taking the time to review our manuscript and provide feedback. We are happy to clarify the novelty further both here and in the revisions to the manuscript text as detailed below. The novelty lies in the first demonstration of an approach of increasing information dimensionality of label-free SERS signatures through the combination of untargeted surface receptor spectral profiles. By implementing this strategy, we address key challenges in the field of label-free SERS and have provided a thorough investigation of the underlying principle and a demonstration of its implicit value with a controlled set of examples. This manuscript provides a strategy of using the rich chemical and structural information inherently present in SERS signatures as input signals for an artificial-nose empowered sensor. We believe our work will be of great interest to the community (as recognised by Reviewers 1 and 2). We hope the responses detailed below will adequately address the reviewer's points and clarify the novelty of the work.

My concerns are given below

1. Label free SERS is only applicable when the target molecule has inherently large Raman cross section. However this is not true for many biologically relevant cases for small molecules (such as glucose) or biomarker detection (proteins). So by doing FASERS (basically nothing advanced but a modified Self assembled monolayer/multilayer (SAM), the fundamental issue with label free SERS could not overcome to a great extent.

The reviewer is correct in stating that many biologically relevant molecules may arguably present small Raman cross-sections and are therefore difficult to detect. We also fully agree with the reviewer that this is indeed one of the challenges that state-of-the-art label-free SERS research has endeavoured to overcome. In addition however, practical challenges of the label-free approach arise from several areas: molecules with the largest Raman cross-section tending to dominate the signal, compositional diversity of biomolecules yielding highly overlapping spectra, and the analyte of interest being outcompeted by the binding of abundant components of biological systems. Our approach endeavours to address all of these challenges through its design.

Importantly, we respectfully disagree with the reviewer's perspective that label-free SERS, as a result of the low Raman cross-section of certain biological molecules, is therefore not an appropriate tool for biologically relevant molecules. Raman characterisation methods represent an established approach for discrimination in biological systems. Such approaches are indeed limited by low Raman cross-section biomolecules being omitted from analysis. In this context, one of the promises of SERS is in its ability to amplify the response to low analyte concentrations or weakly scattering molecules. Label-free SERS seeks to retain the merits of Raman spectroscopic methods (ensemble analysis) at an amplified, surface-confined environment. This allows the surface composition to determine the biomolecules that are interrogated, and very importantly, their orientation and location with respect to the plasmonic surfaces. This enables the opportunities we have capitalised upon in this work to tailor the functional environment of the SERS surfaces and enable ensemble biomolecule measurements dramatically increasing the data available to inform PCA-LDA methods. We would like to point out that label-free SERS has shown promise not only as a detection tool, on which the reviewer has focussed, but also a biochemical analysis technology in various biomedical fields, ranging from detection of biomolecules^{12, 13, 14, 15, 16, 17, 18, 19}, drug monitoring²⁰,

^{21, 22, 23}, studies of molecular structures of biomolecules^{24, 25, 26, 27}, identifying biological species^{28, 29, 30, 31, 32}, diagnosing diseases^{33, 34, 35}, through to monitoring biological processes at the cellular level^{36, 37, 38, 39}.

We appreciate the reviewer's comment highlights the need for improved clarity in the manuscript in explaining the promise of label-free research and have therefore made the following changes to the main text:

Revision made:

- **Text in the manuscript page 3** has been revised accordingly:

There are huge benefits to be had in moving towards platform diagnostic technologies that are not reliant on target-specific binding structures (antibodies, aptamers etc.) and the associated burden of their discovery, complex conjugation and production procedures. A number of targeting-free sensing technologies are being developed that seek to meet this goal^{40, 41, 42}, and among them, label-free surface-enhanced Raman spectroscopy (SERS) has attracted considerable attention with the promise of sensitive direct profiling of the unique fingerprints of a biological sample in a wash- and label-free format^{43, 44, 45}. Compared to a targeted approach of detecting the presence of a specific analyte by measuring signals from pre-tagged Raman reporters, a key advantage of label-free SERS is that it is not necessarily limited to pre-specification of targets of interest and the challenge of developing targeting molecules and labelling them with SERS-tags. Label-free SERS has therefore been particularly successful where target-binding entities have not yet been established, and where spectral information can inform of changes in molecular structure. There is great value when the compositional diversity of molecules that is encoded in the unique SERS signatures is interrogated instead of limiting to specific molecules as a target. Indeed, label-free SERS has enabled detection of biomolecules^{12, 13, 14, 15, 17, 19}, drug monitoring^{20, 21, 23}, studies of molecular structures of biomolecules^{25, 26, 27}, identifying biological species^{28, 29, 31, 32}, diagnosing diseases^{33, 34, 35}, through to monitoring biological processes at the cellular level^{36, 37, 39}. Despite the promises that arise from the sensitivity of SERS to molecular orientation and separation from the plasmonic surface, there are limitations due to the inherent chemical and structural complexity of biomolecules that yield overlapping spectra. This is often viewed as an insurmountable challenge necessitating methods to specifically bind target analytes that may otherwise present low Raman scattering cross sections⁴⁶ and weak affinities to the plasmonic surfaces of SERS sensors or may be outcompeted by binding of abundant components of biological systems^{47, 48}.

With respect to the reviewer's comment on the use of self-assembled monolayers (SAM), we believe they can be used to effectively address the limitations of the low Raman cross-section of biological molecules, indeed this is what our manuscript presents. The groundwork for this is described in studies showing the efficacy of SAMs towards improving the selectivity of the surface without employing target-receptor pairs. As we discussed in the manuscript, SAMs have previously been employed with label-free SERS applications to more selectively bind low-scattering cross section molecules such as glucose. We would like to refer to the introduction section detailing the merit of self-assembled monolayers in label-free SERS that reads: *"Self-assembled monolayers (SAMs) have been explored in several different ways to introduce more selectivity towards specific analytes at the plasmonic surface of SERS sensors. For instance, zwitterionic SAMs can resist nonspecific fouling of proteins in complex media and minimize their contribution to SERS spectra^{2, 48}. Combinations of SAMs have been tailored*

to improve selectivity towards certain small molecules such as glucose^{49, 50, 51, 52, 53} and 3,4-methylenedioxymethamphetamine (MDMA)⁵⁴”.

In this regard, we would like to emphasize that FASERS is not merely another example of an optimized SAM for the detection of a pre-selected specific analyte (e.g. drugs, glucose). Our work deviates significantly from the concept of improved focussing of selected target molecules toward quantitative detection, and also from the concept of minimizing the off-target biological system components (e.g. adsorbed proteins). Instead, we move towards diversifying and parallelising the low-specificity spectral data to amplify the sample information obtainable from surface-dependent spectral signature modulation. We present the merit of the strategy of combining untargeted SAM spectral signatures from an arrayed sensor enabling effective discrimination by increasing information dimensionality. We highlight that this approach helps circumvent another hurdle of variability in spectral signatures by label free SERS. We believe that the obvious variation observed in SERS signals from model systems and the study of underlying mechanism by molecular dynamics simulation clearly establishes the principle of this concept.

We appreciate the reviewer’s comment highlights the need for improved clarity in the manuscript in defining the novelty of our work and have therefore made the following changes to the main text.

Revision made:

- **Text in the manuscript page 3–5** has been revised accordingly:

Self-assembled monolayers (SAMs) have been explored in several different ways to introduce more selectivity towards specific analytes at the plasmonic surface of SERS sensors. For instance, zwitterionic SAMs can resist nonspecific fouling of proteins in complex media and minimize their contribution to SERS spectra^{2, 48}. Combinations of SAMs have been tailored to improve selectivity towards certain small molecules **with low Raman scattering cross section** such as glucose^{49, 50, 51, 52, 53} and 3,4-methylenedioxymethamphetamine (MDMA)⁵⁴. These prior studies have focused on the effective optimization of the SAM composition to improve selectivity without a specific receptor^{49, 50, 54}, or to minimize the influence of off-target biological system components to enhance the analyte signal^{2, 48}. Such approaches aim to reduce the complexity of SERS fingerprints. However, there is significant potential in general methods that can capitalize upon the rich compositional information present within the overlapping spectra rather than attempting to minimize it. Artificial-nose approaches represent a promising strategy towards embracing the compositional diversity when interrogating biological samples^{55, 56, 57}. **In such approaches, low-specificity physicochemical interactions at arrays of different sensing receptors are used, each yielding physical signals that can be recorded and combined to generate a patterned output. The effectiveness of this addition of data dimensionality from an array of output channels is assisted by chemometric data analysis to build classifiers towards hypothesis-free sample identification. These approaches however, frequently monitor 1-dimensional outputs such as fluorescence intensity^{58, 59}, electrical response⁶⁰ or mass change⁶¹ per detection receptor. Label-free SERS in contrast offers the potential for 2-dimensional readouts per receptor drastically amplifying the amount of chemical, structural information obtainable for a small arrayed sensor system.**

We demonstrate that it is possible to substantially enhance the accuracy of biological sample identification without target-specific binding receptors by increasing data dimensionality through multiple SAM-functionalized surfaces while embracing the spectral complexity of the label-free SERS datasets. The approach is employed as an arrayed sensing platform, which we term “Functionalized Array for Surface-Enhanced

Raman Spectroscopy (FASERS)”, consisting of 8 different SAMs formed on plasmonic Au-nanopillars. A series of un-targeted, mildly-selective SAMs have been employed in our system to promote diverse ranges of physico-chemical interactions with different sample constituents rather than improving the detection of pre-selected target molecules or minimizing the off-target biological system components. Through selective SERS enhancement of molecules in close proximity to each surface, we sought to present the diversified SERS signatures detectable in complex liquids. In this approach, the compositional diversity of biologically-derived liquids, which has previously limited conventional label-free SERS biosensors, is leveraged by introducing another dimensionality to the obtained datasets. We illustrate these varied interactions by assessing the binding of 4 small molecules to the SAMs of differing molecular characteristics and explore the manifold range of interactions at play using molecular dynamics simulations. The SAM-dependent multi-dimensional spectral datasets can identify and discriminate complex biological samples with higher accuracy compared to conventional label-free SERS. The merit of this approach is shown for two cell lysates from companion cell lines that comprise malignant and normal human cell lines from the same tissue of a patient, where by combining the spectral signatures for multiple SAMs, the classification accuracy of label-free SERS can be improved in a facile manner. Our approach highlights that by exploiting label-free SERS and its ability to interrogate biomolecules as a function of SERS sensor surface functionality, a powerful artificial-nose empowered strategy can be envisaged.

2. Most significantly, authors adopted FASERS to differentiate the breast carcinoma cells with normal fibroblast-like cells from the same individual by employing established statistical tool like PCA/PCA-LDA. There is no advancement in this aspect of the work. First of all, authors should have studied large sets of samples that has statistical significance (at least 20-30 cancer and may be same number of healthy samples). Comparing the data from 1 set of samples is not acceptable and wont convince any. There are many reports on simple label free SERS with statistical classification that resulted in high accuracy (sensitivity/specificity) in the range of over 90%. (some of these works are listed below). In this context, the advancement made using FASERS is quite minimal.

i) *Biosensors and Bioelectronics* 25, 2414–2419 (2010)

ii) *Analyst*, 138, 3967 (2013)

iii) *Optics Express* 19 (14), 13565-13577, (2011)

Differentiating normal and cancer cells (lysate) is far easier than differentiating different types of closely associated cancers. If FASERS provides high specificity, authors should have demonstrated it to establish its superiority. As indicated in above references, differentiating cancer Vs Normal directly from patient’s serum (or body fluids) is significant and has much more clinical value than testing it in the cell lysate as demonstrated here. With FASERS approach, I assume it is really difficult to achieve highly sensitive detection in a real complex body fluids (such as blood or saliva) by overcoming the biofouling. Authors should have demonstrated it in this paper.

We wholeheartedly agree with the reviewer that presenting a diagnostic as capable of differentiating cancer types requires a larger scale evaluation of clinical samples. However, we would like to point out that it is not our intention to present this as a diagnostic technology at this stage and we believe that the data we have shown are sufficient to support the viability of our concept in this proof-of-principle. We have selected these samples as exemplary systems where we can effectively control for sample handling variation (controlling for this in a clinical study would require large sample size, as well as negative controls (both healthy and diseased). We believe we have been able to demonstrate the merit of the FASERS system

with controlled lysate samples that have been handled in identical manners to ensure differences are the results of biological composition rather than handling/storage or other biases.

The reviewer correctly highlights several examples of label-free SERS being used for discrimination of cancerous from non-cancerous clinical samples. However, we would like to point out that the focus of our study deviates from showing the effective use of SERS as a tool for cancer diagnostics. Instead, we propose the novelty of our work lies in the promises of amplifying the information through modulation of SERS enhancement of constituent molecules from differently interacting surfaces. The implicit value of this arrayed approach is in that we can construct a classifier of improved accuracy not by increasing the number of training sets but by increasing information dimensionality obtainable from each sample. We believe this has been well supported in that our mean discriminatory accuracy improves with an increase in the number of SAMs employed (Figure 5c).

We agree the FASERS will hold a clinical value as a diagnostics tool only through extensive clinical evaluation, but we also would like to point out that performing such a clinical evaluation would require a prolonged and thorough study that would in itself warrant separate publication. It would require, particularly for a hypothesis-free diagnostic, ensuring that the results indicate the presence of different pathologies rather than other biases such as sample collection, storage or comorbidities or correlated variables (fever, dietary etc.) which is beyond the scope of this already in-depth study. In the manuscript's current form, we endeavored to assess matching matrices (cell-lysates), cultured and handled in identical manners to provide confidence that we were observing biological differences between the cell-lines in this study. This has allowed us to achieve the main purposes of this study: to introduce the concept of the FASERS using controlled yet complex sample systems.

While we do not believe it is necessary to perform such clinical evaluations in this manuscript, we acknowledge the reviewer's comment and we clarify that this is one of our future research directions, and have therefore made the following changes to the main text.

Revision made:

- **Text in the manuscript page 22** has been revised accordingly:

This implementation of an artificial-nose empowered SERS approach to generate and analyze high dimensionality data by increasing the number of spectral outputs from a single sample represents a proof-of-concept artificial-nose based sensing methodology. Extending this initial demonstration of the FASERS platform to other biologically relevant systems is the next step in evaluating the potential for sample characterization and discrimination. **Particularly, an in-depth clinical study with patient-derived biological fluids such as blood plasma represents an important follow-up research direction towards a hypothesis-free diagnostics platform. We propose this could be realized by careful selection of SAMs and automated methods to collect, statistically analyze, and interpret these data sets such as artificial-intelligence-based feature selection. When expanding to clinical-derived systems with larger sample sizes, this would enable optimal model parsimony to reduce aberrant noise and other spurious signatures that do not contribute to sample discrimination.** This offers a new platform technology to characterize, cluster and classify biological samples, which in the context of diseases, such as cancer, opens up numerous possibilities for a simple yet powerful label-free diagnostic platform.

With respect to the reviewers concerns regarding biofouling, this is indeed a common challenge when moving to patient-derived samples. The SAMs employed in this work are not

intended to be optimized to reduce biofouling. They are shown to be effective monolayers for the analysis of cell-lysates and assessing the fundamental interactions at play for small controlled molecules. Moving to plasma or serum, we would expect the underlying principle of FASERS to add value in label-free discrimination, however the concept could be expanded to incorporate well-established SAM compositions that specifically aim to reduce biofouling in clinically derived samples. However, this does not diminish the results presented in this work that effectively underlie the principle of FASERS using cell-lysates.

The reviewer also commented on the advancement in the statistical analysis. Indeed, our fundamental principle of statistical analysis is based on the well-established PCA-LDA tool, however we have developed a non-conventional approach by cross-combining principal components (PC) obtained from multiple SAMs. This clearly deviates from the conventional approach of increasing dimensionality by employing several PCs (e.g. PC1, PC2, PC3, PC4 *etc.*) from the same data set. We have utilized the first PC (PC1) from each SAM for our modified analysis to ensure that we are exploiting only the largest variability from each data set. We have gradually increased the number of SAMs included in the analysis to show the merit of amplifying the dimensionality of information, demonstrated by increase of mean discriminatory accuracy as a function of the number of SAMs used. We believe we have validated our approach by comparing the position of these largest variabilities with the tentative assignments of biomolecules (Figure 5c) and by performing leave-one-out cross validation on all of the constructed classifiers as shown in figure 5e and Supplementary Information Figure S11.

We appreciate the reviewer's comment highlights the need for improved clarity in the manuscript in explaining our statistical approaches and have therefore made the following changes to the main text.

Revision made:

- **Text in the manuscript page 20** has been revised accordingly:

Figure 5f shows a scatter plot of the first two PCs for the normal and cancerous cell lysates on the non-functionalized Au-nanopillars. The corresponding accuracy of the discrimination was 75% and a significant overlap of the clusters from the two cell lysates was observed. The implication is that the spectral differences between the two samples were not sufficient to fully discriminate using the non-functionalized substrates alone. **Inspired by an artificial-nose-based approach,** we performed a non-conventional PCA-LDA where the PC1s from **multiple SAMs** were cross-combined for analysis. **In this approach, instead of increasing dimensionality by employing several PCs (e.g. PC1, PC2, PC3, PC4 *etc.*) from the same data set as in conventional PCA-LDA, we have utilized the first PC (PC1) from multiple SAMs and gradually increased the number of SAMs included in the analysis to show the merit of amplifying the dimensionality of information.**

3) *For any practical applications, specifically for bio medical, if the analyte of interest has weak Raman cross section, targeted detection (using targeting moieties like antibodies/aptamers or using SERS nanotag with a reporter molecule) is always well accepted and had shown highly promising results. In this context, the label free approach with “self-assembled layer” is not innovative or revolutionary.*

As we have described in the Reviewer 3 Question 1 response in detail, label-free SERS seeks to retain the merits of Raman spectroscopic methods (ensemble analysis) where the biochemical and structural information of molecules are interrogated. This differs considerably from targeted approaches which can be very effective if biomarkers are known and affinity reagents well established, but such approaches are not suited to exploit the potential of ensemble measurements where it is variations in ensembles of biomarkers that may hold the discriminatory value. Label-free SERS generates spectral signatures containing the chemical and structural information of the analytes of interest, rather than measuring signals from pre-tagged Raman reporter molecules for indirectly detecting the presence of specific analytes as employed in a targeted SERS approach. A key advantage of label-free SERS is that it is not necessarily limited to pre-specification of targets of interest and the challenge of developing targeting molecules and their labelling with SERS-tags. Label-free SERS has been therefore particularly successful where target-binding entities (e.g. antibodies, aptamers) have not yet been established (e.g. drugs) and where spectral information can inform of changes in molecular structure. There is great value when the compositional diversity of molecules that is encoded in the unique SERS signatures is interrogated instead of limiting to specific molecules as a target.

Given the strong advantages of label-free SERS, we believe new methodologies that can advance label-free SERS hold great value in biomedical fields both as detection and analytic technologies. This is clearly demonstrated by significant interest and recent advances in the field as detailed in the Reviewer 3 Question 1 response¹²⁻³⁹.

In terms of the novelty of using the self-assembled monolayer (SAM) in our work, we would like to emphasize our novelty lies not in the idea of using of SAM to promote specific target detection, but in employing an array of differently modified surfaces as multiple output channels to diversify our signals. We believe our work clearly deviates from previous SAM-employed studies as it pursues a hypothesis-free label free methodology that does not specify any analytes of interest, but rather promote diverse low-specificity interaction of the constituent molecules in the complex liquid media that are therefore profiled through modulated signatures and comprehensively interpreted. To the best of our knowledge, we have for the first time demonstrated to utility of the amplified information dimensionality of SERS in a label- and wash-free format via low-specificity interactions between surface and complex media. We believe the we have sufficiently demonstrated the underlying mechanism of SERS modulation at each SAM and its inherent value of increasing the number of spectral-output channels employed.

4-1) *As reported, if this work has the advancement as “artificial-nose based sensing methodology”, authors should have demonstrated it in a various complex media that include biological/non-biological.*

We find the reviewer’s suggestion of testing biological and non-biological complex samples is rather ambiguous. We would like to point out that the fundamental aim of artificial-nose based sensing lies in the idea of increasing data dimensionality by generating a patterned output through multiple output channels^{55, 56, 57}. Rather than evaluating multiple complex sample systems, we would like to clarify that we chose instead to investigate single molecular solutions (i.e. p-phenylenediamine, 4-aminophenylacetic acid, folic acid and rhodamine 6G) as controlled analytes. The rationale behind choosing these controlled matrices is to clearly focus on the correlation of underlying interactions and resulting differences in spectral

signatures, that otherwise may not have been sufficiently distinguishable due to overlapped signatures. By using these controlled matrices, we were able to understand the diversified SERS signatures of the same molecule on each surface, arising from the modulation of interactions between the molecule and SAMs of distinct molecular characteristics. Correlating these results with the molecular dynamics studies, we were able to sufficiently support our methodology of utilising varied SAM functionalizations to generate different output channels (Figure 3, 4).

We agree with the reviewer that demonstration with different classes of biological systems where SERS signals can be modulated is of interest and is a future research target for us. We believe however that our demonstration with well controlled cell lysate samples (as described in reviewer 3 response 2) stands as a sufficient demonstration of the FASERS approach. However, to show the versatility of the FASERS approach in generating diverse spectral signatures of complex samples, we have added initial data for purified extracellular vesicles (EVs) isolated from MDA-MB-231 breast cancer cells as an additional model biological system. EVs differ from cell lysates where biomolecules (e.g. proteins, lipids, nucleic acids, glycogens, metabolites) exhibiting compositional diversity do not exist freely in liquid media and are instead vesicular entities with complex surface composition. EVs used in the analysis have been purified by size exclusion chromatography to remove soluble protein and other components present in the secretome, to ensure the label-free SERS signatures from the solution can to be attributed to the variation of EV composition and orientation near the surface rather than artefacts from other impurities. Introducing a more complex sample system following the small molecule studies, the data has now been discussed in the main text and Supplementary Information as follows:

Revision made:

- **Text in the manuscript page 16** has been revised accordingly:

Unlike those of single component molecule solutions, the SERS spectra of biological samples are expected to be more complex resulting from the overlapped signatures of a myriad of biomolecules, spanning proteins, lipids, nucleic acids, glycogens and metabolites. As a demonstration of a compositionally complex, biologically derived system, but in controlled, purified media, we employed FASERS for investigation of extracellular vesicles (EVs) isolated from MDA-MB-231 human breast cancer cells (Supplementary Information, Figure S8). EVs used in the analysis were purified by size exclusion chromatography to remove soluble protein and other components present in the secretome, to ensure the label-free SERS signatures from the solution could be attributed to the variation of EV composition and orientation near the surface rather than artefacts from other impurities. Peak positions, shapes and intensities within the EV fingerprints corresponding to proteins, lipids, and nucleic acids varied significantly depending on SAM functionality. The implication of this is that the EVs were interacting at the gold or SAM surfaces enabling the SERS active regions to survey the EV composition near the surface. Alternatively, it could reflect a degree of dissociation and release of their contents during interaction with different SAMs. Either way, the variation observed demonstrates the merit of multiple SAM output channels in a controlled complex sample.

Having established the feasibility of diverse signal generation using FASERS in a complex yet controlled EV biological system, we employed an artificial-nose empowered approach to validate how obtaining spectral data as a function of sensor surface chemical functionality can improve discrimination of complex biological samples. Although protein is one of the key elements in the biological samples, the adsorption of the most abundant proteins often leads to saturation of most of the SERS hot spots and can hinder other small biomolecules from approaching².

- Figure S8 in the Supplementary Information section IV has been added:

Figure S8. Series of SERS spectra from purified extracellular vesicles (EVs) from MDA-MB-231 human breast cancer cells obtained from functionalized Au-nanopillars with various SAM-forming molecules. Each monolayer is referred to as: non-functionalized (bare), 1-propanethiol (3CH₃), 3-mercapto-1-propanol (3OH), 3-mercaptopropionic acid (3COOH), 3-amino-1-propanethiol (3NH₂), 1-undecanethiol (11CH₃), 11-mercapto-1-undecanol (11OH), 11-mercaptopundecanoic acid (11COOH), 11-amino-1-undecanethiol (11NH₂).

Supported by this additional EV data, we believe our above responses to the reviewer highlight the novelty of our work and the viability of the technique—the design and effectiveness of the platform technology—sufficiently.

4-2) Also, establishing the limit of detection that breaks the current limit for some of the trace biochemicals would have attracted great interest. Even establishing the detection of glucose in a physiologically relevant condition should have attracted interest.

The reviewer has raised an important point regarding the limit of detection, which is a frequent concern in detection technologies for trace biochemicals. Nonetheless, we would like to respectfully point out that since our methodology does not aim for specific target detection nor for more sensitive design of SERS substrates, breaking the current limit of detection is out of the scope of our work. As described in detail in the Reviewer 3 Question 1–4 responses, our system proposes a hypothesis-free label-free methodology where it is not intended to detect any pre-selected biochemical molecules (e.g. glucose), although that is not to say it would not be possible to. As discussed in earlier responses however, detection of biochemicals (e.g. glucose) has already been demonstrated in label-free SERS using SAMs of optimized compositions as we reference in the manuscript^{49, 50, 51, 52, 53} and was therefore not a target for us.

5) SERS substrate reported here is also not novel. There are many reports on similar SERS substrates. It would have been nice if the SERS mapping is demonstrated to establish the reproducibility in signal at various spatial location of the substrate.

We thank the reviewer for their feedback. We agree with the reviewer that there have been intensive efforts to develop novel designs of SERS substrates enabling sensitive and reproducible SERS measurements.

Nonetheless, as we have described in the Reviewer 3 Question 4 in detail, the aim and the novelty of our work does not reside with comparing sensitivity of the SERS substrates with those previously reported. Instead, our novelty is in the approach of utilising the diverse range of low-specificity interactions between the SAM and the constituent molecules to diversify the outputs of the signatures, which enable higher dimensionality fingerprinting. Thus, whilst we have utilised a specific design of substrates—gold film coated Si_3N_4 nanopillars, we do not claim novelty in this, instead employing them as a reliable substrate for the investigation. They are derived from systematic optimization of the structure design and fabrication procedures within our research group.

With respect to the reproducibility of our signals at various spatial location of the substrates, the data used in the demonstration of FASERS uses spectra generated at different regions of each SERS substrate thus this variability is contained within the raw data associated with the manuscript. Thus, the low level of signal variations represented by small standard deviations (i.e. shaded area in Figure 3b–e, error bars in Figure 3f–i) have demonstrated our reliability of SERS substrates. However, we appreciate the reviewer's suggestion that the SERS mapping will further establish the spatial reproducibility in signal, and have performed SERS mapping ($50\ \mu\text{m} \times 50\ \mu\text{m}$, pixel size of $1\ \mu\text{m}^2$) of chemisorbed $1\ \mu\text{M}$ of 4-mercaptobenzoic acid (4-MBA) on four randomly-selected spatial locations on the Au nanopillar substrate (Au nP). We have observed significant SERS enhancement of 4-MBA signals across the entire scanned area compared to a Flat Au surface with marginal signal variation. Importantly, we have observed no/little apparent mean signal variation between the four different spatial location, indicating a good spatial reproducibility of the enhancement. The data has now been discussed in the main text and Supplementary Information as follows:

Revision made:

- **Text in the manuscript page 8** has been revised accordingly:

We fabricated SERS-active gold film coated nanopillar substrates (Au-nanopillars) via colloidal lithography and plasma etching as described in Figure 2a and Figure S1a–b (Supplementary Information). The peak intensities increased with the increasing ATP concentration, up to a saturation point that likely correlated with complete coverage of the SERS active surface (Figure 2b, Figure S1c, Supplementary Information). **To establish the spatial reproducibility of the SERS enhancement, we performed SERS mapping on Au-nanopillars substrates using the prominent peak area ($1057.1\text{--}1091.7\ \text{cm}^{-1}$), centered at $1074.7\ \text{cm}^{-1}$, of chemisorbed $1\ \mu\text{M}$ 4-mercaptobenzoic acid (4-MBA) (Figure S2, Supplementary Information). Significant SERS enhancement of 4-MBA signals across the entire scanned area was observed compared to those of a flat gold film-coated surface (Au-Si). Importantly, we observed little to no significant signal variation between the different spatial locations, indicating the robust spatial reproducibility of the SERS substrate.** Detailed tentative assignments of the peaks are further discussed in Supplementary Information Table S1.

- Figure S2 in the Supplementary Information section II.1. has been added:

II.1. SERS mapping of the Au-nanopillars substrates

Figure S2. SERS mapping of chemisorbed 1 μM 4-mercaptobenzoic acid (4-MBA) on Au-nanopillars and gold film coated Si wafer (Au-Si) substrate. Four randomly-selected spatial locations on the Au-nanopillar substrate (Au nP Site 1–4) were investigated and compared to one location on a Au-Si substrate for control. (a–e) Optical microscope image of (a) Au-Si, (b) Au nP Site 1, (c) Au nP Site 2, (d) Au nP Site 3, (e) Au nP Site 4. (f–j) SERS mapping (50 μm x 50 μm , pixel size of 1 μm^2 , peak at 1074.7 cm^{-1}) of corresponding location: (f) Au-Si, (g) Au nP Site 1, (h) Au nP Site 2, (i) Au nP Site 3, (j) Au nP Site 4. Scale bar represents 10 μm . Each spectrum was smoothed, baseline subtracted and normalized by the area under the curve. The maximum value of the intensity bar corresponds to the maximum intensity out of all the measurements performed on all of the different sites. (k) Normalized SERS intensity of peak at each pixel (50 x 50) of the scan. Significant SERS enhancement of 4-MBA signals across the entire scanned area were observed compared to those of Flat Au surface (Au-Si). Importantly, we have observed no/little apparent mean signal variation between the different spatial location, indicating a good spatial reproducibility of the enhancement

Correction of Axis Labelling for Figure S10

While revising the manuscript we noted incorrect axis labelling of Figure S10 (Supplementary Information, page 21). This has been corrected as follows with correct percentage of variance.

Figure S10. Three-dimensional PCA scatter plots for Hs578Bst (normal) and Hs578T (cancerous) cell lysates using multiple SAM-functionalization. (I–XVI) Representative 3D scatter plots from combinations of different SAM-functionalizations that represent 100% accuracy in cancerous cell lysates discrimination. Each SAM is referred to as: non-functionalized (bare), 1-propanethiol (3CH₃), 3-mercapto-1-propanol (3OH), 3-mercaptopropionic acid (3COOH), 3-amino-1-propanethiol (3NH₂), 1-undecanethiol (11CH₃), 11-mercapto-1-undecanol (11OH), 11-mercaptoundecanoic acid (11COOH), 11-amino-1-undecanethiol (11NH₂). The blue planes depicting classification were derived using linear discriminant analysis (LDA) as the classification algorithm to separate the two groups. The red and green ellipsoids represent one standard error of each group. Insets are the calculated accuracy in cancer cell lysates discrimination.

References

1. Ruddon, R. W. What makes a cancer cell a cancer cell? in *Holland-Frei Cancer Medicine* (eds. Kufe D. W., et al.). 6th Edition, Chapter 9. (B. C. Decker, 2003).
2. Sun, F. *et al.* Hierarchical zwitterionic modification of a SERS substrate enables real-time drug monitoring in blood plasma. *Nat. Commun.* **7**, 13437 (2016).
3. Bryant, M. A. & Pemberton, J. E. Surface Raman scattering of self-assembled monolayers formed from 1-alkanethiols: behavior of films at gold and comparison to films at silver. *J. Am. Chem. Soc.* **113**, 8284–8293 (1991).
4. Kudelski, A. Structures of monolayers formed from different HS—(CH₂)₂—X thiols on gold, silver and copper: comparative studies by surface-enhanced Raman scattering. *J. Raman Spectrosc.* **34**, 853–862 (2003).
5. Bryant, M. A. & Pemberton, J. E. Surface Raman scattering of self-assembled monolayers formed from 1-alkanethiols at silver [electrodes]. *J. Am. Chem. Soc.* **113**, 3629–3637 (1991).
6. Sandhyarani, N. & Pradeep, T. Characteristics of alkanethiol self assembled monolayers prepared on sputtered gold films: a surface enhanced Raman spectroscopic investigation. *Vacuum* **49**, 279–284 (1998).
7. Michota, A., Kudelski, A. & Bukowska, J. Chemisorption of cysteamine on silver studied by surface-enhanced Raman scattering. *Langmuir* **16**, 10236–10242 (2000).
8. Michota, A., Kudelski, A. & Bukowska, J. Influence of electrolytes on the structure of cysteamine monolayer on silver studied by surface-enhanced Raman scattering. *J. Raman Spectrosc.* **32**, 345–350 (2001).
9. Michota, A., Kudelski, A. & Bukowska, J. Molecular structure of cysteamine monolayers on silver and gold substrates: comparative studies by surface-enhanced Raman scattering. *Surf. Sci.* **502–503**, 214–218 (2002).
10. Kudelski, A. Raman study on the structure of 3-mercaptopropionic acid monolayers on silver. *Surf. Sci.* **502–503**, 219–223 (2002).
11. Królikowska, A., Kudelski, A., Michota, A. & Bukowska, J. SERS studies on the structure of thioglycolic acid monolayers on silver and gold. *Surf. Sci.* **532–535**, 227–232 (2003).
12. Xu, L.-J., Zong, C., Zheng, X.-S., Hu, P., Feng, J.-M. & Ren, B. Label-free detection of native proteins by surface-enhanced Raman spectroscopy using iodide-modified nanoparticles. *Anal. Chem.* **86**, 2238–2245 (2014).
13. Haynes, C. L., Yonzon, C. R., Zhang, X. & Van Duyne, R. P. Surface-enhanced Raman sensors: early history and the development of sensors for quantitative biowarfare agent and glucose detection. *J. Raman Spectrosc.* **36**, 471–484 (2005).
14. Ranc, V. *et al.* Magnetically assisted surface-enhanced Raman scattering selective determination of dopamine in an artificial cerebrospinal fluid and a mouse striatum using Fe₃O₄/Ag nanocomposite. *Anal. Chem.* **86**, 2939–2946 (2014).
15. De Luca, A. C., Reader-Harris, P., Mazilu, M., Mariggìò, S., Corda, D. & Di Falco, A. Reproducible surface-enhanced Raman quantification of biomarkers in multicomponent mixtures. *ACS Nano* **8**, 2575–2583 (2014).

16. Torres-Nuñez, A., Faulds, K., Graham, D., Alvarez-Puebla, R. A. & Guerrini, L. Silver colloids as plasmonic substrates for direct label-free surface-enhanced Raman scattering analysis of DNA. *Analyst* **141**, 5170–5180 (2016).
17. Morla-Folch, J. *et al.* Ultrasensitive direct quantification of nucleobase modifications in DNA by surface-enhanced Raman scattering: the case of cytosine. *Angew. Chem., Int. Ed.* **54**, 13650–13654 (2015).
18. Morla-Folch, J., Alvarez-Puebla, R. A. & Guerrini, L. Direct quantification of DNA base composition by surface-enhanced Raman scattering spectroscopy. *J. Phys. Chem. Lett.* **7**, 3037–3041 (2016).
19. Yüksel, S., Schwenkbier, L., Pollok, S., Weber, K., Cialla-May, D. & Popp, J. Label-free detection of *Phytophthora ramorum* using surface-enhanced Raman spectroscopy. *Analyst* **140**, 7254–7262 (2015).
20. Hidi, I. J., Jahn, M., Pletz, M. W., Weber, K., Cialla-May, D. & Popp, J. Toward levofloxacin monitoring in human urine samples by employing the LoC-SERS technique. *J. Phys. Chem. C* **120**, 20613–20623 (2016).
21. Hidi, I. J. *et al.* Lab-on-a-chip-surface enhanced Raman scattering combined with the standard addition method: toward the quantification of nitroxoline in spiked human urine samples. *Anal. Chem.* **88**, 9173–9180 (2016).
22. Subaihi, A. *et al.* Quantitative online liquid chromatography–surface-enhanced Raman scattering (LC-SERS) of methotrexate and its major metabolites. *Anal. Chem.* **89**, 6702–6709 (2017).
23. Berger, A. G., Restaino, S. M. & White, I. M. Vertical-flow paper SERS system for therapeutic drug monitoring of flucytosine in serum. *Anal. Chim. Acta.* **949**, 59–66 (2017).
24. Buividas, R. *et al.* Statistically quantified measurement of an Alzheimer's marker by surface-enhanced Raman scattering. *J. Biophotonics* **8**, 567–574 (2015).
25. Reymond-Laruinaz, S., Saviot, L., Potin, V. & Marco de Lucas, MdC. Protein–nanoparticle interaction in bioconjugated silver nanoparticles: a transmission electron microscopy and surface enhanced Raman spectroscopy study. *Appl. Surf. Sci.* **389**, 17–24 (2016).
26. Mu, Z., Zhao, X., Huang, Y., Lu, M. & Gu, Z. Photonic crystal hydrogel enhanced plasmonic staining for multiplexed protein analysis. *Small* **11**, 6036–6043 (2015).
27. Miljanić, S., Ratkaj, M., Matković, M., Piantanida, I., Gratteri, P. & Bazzicalupi, C. Assessment of human telomeric G-quadruplex structures using surface-enhanced Raman spectroscopy. *Anal. Bioanal. Chem.* **409**, 2285–2295 (2017).
28. Yan, B. & Reinhard, B. M. Identification of tumor cells through spectroscopic profiling of the cellular surface chemistry. *J. Phys. Chem. Lett.* **1**, 1595–1598 (2010).
29. Tang, H-W., Yang, X. B., Kirkham, J., Smith, D. A. Probing intrinsic and extrinsic components in single osteosarcoma cells by near-infrared surface-enhanced Raman scattering. *Anal. Chem.* **79**, 3646–3653 (2007).

30. Jarvis, R. M., Brooker, A. & Goodacre, R. Surface-enhanced Raman spectroscopy for bacterial discrimination utilizing a scanning electron microscope with a Raman spectroscopy interface. *Anal. Chem.* **76**, 5198–5202 (2004).
31. Premasiri, W. R., Moir, D. T., Klempner, M. S., Krieger, N., Jones, G. & Ziegler, L. D. Characterization of the surface enhanced Raman scattering (SERS) of bacteria. *J. Phys. Chem. B* **109**, 312–320 (2005).
32. Patel, I. S., Premasiri, W. R., Moir, D. T. & Ziegler, L. D. Barcoding bacterial cells: a SERS-based methodology for pathogen identification. *J. Raman Spectrosc.* **39**, 1660–1672 (2008).
33. Feng, S. *et al.* Nasopharyngeal cancer detection based on blood plasma surface-enhanced Raman spectroscopy and multivariate analysis. *Biosens. Bioelectron.* **25**, 2414–2419 (2010).
34. Li, S. *et al.* Characterization and noninvasive diagnosis of bladder cancer with serum surface enhanced Raman spectroscopy and genetic algorithms. *Sci. Rep.* **5**, 9582 (2015).
35. Lin, D. *et al.* Label-free blood plasma test based on surface-enhanced Raman scattering for tumor stages detection in nasopharyngeal cancer. *Sci. Rep.* **4**, 4751 (2014).
36. El-Said, W. A., Kim, S. U. & Choi, J-W. Monitoring in vitro neural stem cell differentiation based on surface-enhanced Raman spectroscopy using a gold nanostar array. *J. Mater. Chem. C* **3**, 3848–3859 (2015).
37. Huang, K-C., Bando, K., Ando, J., Smith, N.I., Fujita, K. & Kawata, S. 3D SERS (surface enhanced Raman scattering) imaging of intracellular pathways. *Methods* **68**, 348–353 (2014).
38. Büchner, T. *et al.* Relating surface-enhanced Raman scattering signals of cells to gold nanoparticle aggregation as determined by LA-ICP-MS micromapping. *Anal. Bioanal. Chem.* **406**, 7003–7014 (2014).
39. Lussier, F., Brulé, T., Vishwakarma, M., Das, T., Spatz, J. P. & Masson, J-F. Dynamic-SERS optophysiology: a nanosensor for monitoring cell secretion events. *Nano Lett.* **16**, 3866–3871 (2016).
40. Zeng, S., Baillargeat, D., Ho, H-P. & Yong, K-T. Nanomaterials enhanced surface plasmon resonance for biological and chemical sensing applications. *Chem. Soc. Rev.* **43**, 3426–3452 (2014).
41. Poghossian, A. & Schöning, M. J. Label-free sensing of biomolecules with field-effect devices for clinical applications. *Electroanalysis* **26**, 1197–1213 (2014).
42. Cooper, M. A. Label-free screening of bio-molecular interactions. *Anal. Bioanal. Chem.* **377**, 834–842 (2003).
43. Stiles, P. L., Dieringer, J. A., Shah, N. C. & Van Duyne, R. P. Surface-enhanced Raman spectroscopy. *Annu. Rev. Anal. Chem.* **1**, 601–626 (2008).
44. Bonifacio, A., Cervo, S. & Sergo, V. Label-free surface-enhanced Raman spectroscopy of biofluids: fundamental aspects and diagnostic applications. *Anal. Bioanal. Chem.* **407**, 8265–8277 (2015).

45. Zheng, X-S., Jahn, I. J., Weber, K., Cialla-May, D. & Popp, J. Label-free SERS in biological and biomedical applications: recent progress, current challenges and opportunities. *Spectrochim. Acta. A* **197**, 56–77 (2018).
46. Lane, L. A., Qian, X. & Nie, S. SERS Nanoparticles in medicine: from label-free detection to spectroscopic tagging. *Chem. Rev.* **115**, 10489–10529 (2015).
47. Yang, S., Dai, X., Stogin, B. B. & Wong T-S. Ultrasensitive surface-enhanced Raman scattering detection in common fluids. *Proc. Natl. Acad. Sci. U. S. A.* **113**, 268–273 (2016).
48. Sun, F. *et al.* Stealth surface modification of surface-enhanced Raman scattering substrates for sensitive and accurate detection in protein solutions. *ACS Nano* **9**, 2668–2676 (2015).
49. Lyandres, O., Shah, N. C., Yonzon, C. R., Walsh, J. T., Glucksberg, M. R. & Van Duyne RP. Real-time glucose sensing by surface-enhanced Raman spectroscopy in bovine plasma facilitated by a mixed decanethiol/mercaptohexanol partition layer. *Anal. Chem.* **77**, 6134–6139 (2005).
50. Yonzon, C. R., Haynes, C. L., Zhang, X., Walsh, J. T. & Van Duyne, R. P. A glucose biosensor based on surface-enhanced Raman scattering: improved partition layer, temporal stability, reversibility, and resistance to serum protein interference. *Anal. Chem.* **76**, 78–85 (2004).
51. Stuart, D. A. *et al.* In vivo glucose measurement by surface-enhanced Raman spectroscopy. *Anal. Chem.* **78**, 7211–7215 (2006).
52. Stuart, D. A. *et al.* Glucose sensing using near-infrared surface-enhanced Raman spectroscopy: Gold Surfaces, 10-day stability, and improved accuracy. *Anal. Chem.* **77**, 4013–4019 (2005).
53. Shafer-Peltier, K. E., Haynes, C. L., Glucksberg, M. R. & Van Duyne, R. P. Toward a glucose biosensor based on surface-enhanced Raman scattering. *J. Am. Chem. Soc.* **125**, 588–593 (2003).
54. Stewart, A. & Bell, S. E. J. Modification of Ag nanoparticles with mixed thiols for improved SERS detection of poorly adsorbing target molecules: detection of MDMA. *Chem. Commun.* **47**, 4523–4525 (2011).
55. Miranda, O. R., Creran, B. & Rotello, V. M. Array-based sensing with nanoparticles: ‘chemical noses’ for sensing biomolecules and cell surfaces. *Curr. Opin. Chem. Biol.* **14**, 728–736 (2010).
56. Li, Z., Askim, J. R. & Suslick, K. S. The optoelectronic nose: colorimetric and fluorometric sensor arrays. *Chem. Rev.* 231–292 (2019).
57. Albert, K. J. *et al.* Cross-reactive chemical sensor arrays. *Chem. Rev.* **100**, 2595–2626 (2000).
58. You, C-C. *et al.* Detection and identification of proteins using nanoparticle–fluorescent polymer ‘chemical nose’ sensors. *Nat. Nanotechnol.* **2**, 318 (2007).
59. Le, N. D. B. *et al.* Cancer cell discrimination using host–guest “doubled” arrays. *J. Am. Chem. Soc.* **139**, 8008–8012 (2017).

60. Chang, Y. *et al.* Detection of volatile organic compounds by self-assembled monolayer coated sensor array with concentration-independent fingerprints. *Sci. Rep.* **6**, 23970 (2016).
61. Luo, Y. *et al.* Rapid and simultaneous quantification of 4 Urinary proteins by piezoelectric quartz crystal microbalance immunosensor array. *Clin. Chem.* **52**, 2273–2280 (2006).

REVIEWERS' COMMENTS:

Reviewer #1 (Remarks to the Author):

The authors have fully addressed the comments from all reviewers and I would recommend for the publication on Nature Communications.

Reviewer #2 (Remarks to the Author):

The authors have addressed all my concerns and the paper should be published in its current form.

Editorial Note: In comments to the editor reviewer #3 approved publication after a few small modifications